# ELASTIC FEATURE CONSOLIDATION FOR COLD START EXEMPLAR-FREE INCREMENTAL LEARNING

**Simone Magistri**[1][*]**Tomaso Trinci**[1]**, Albin Soutif–Cormerais**[2]**,**
**Joost van de Weijer**[2]**, Andrew D. Bagdanov**[1]

Department of Information Engineering, University of Florence, Italy[1]
Computer Vision Center, Universitat Autònoma de Barcelona, Spain[2]
`{simone.magistri, tomaso.trinci, andrew.bagdanov}@unifi.it`
`{albin, joost}@cvc.uab.cat`

## ABSTRACT

Exemplar-Free Class Incremental Learning (EFCIL) aims to learn from a sequence of tasks without having access to previous task data. In this paper, we consider the challenging Cold Start scenario in which insufficient data is available in the first task to learn a high-quality backbone. This is especially challenging for EFCIL since it requires high plasticity, which results in feature drift which is difficult to compensate for in the exemplar-free setting. To address this problem, we propose a simple and effective approach that consolidates feature representations by regularizing drift in directions highly relevant to previous tasks and employs prototypes to reduce task-recency bias. Our method, called Elastic Feature Consolidation (EFC), exploits a tractable second-order approximation of feature drift based on an Empirical Feature Matrix (EFM). The EFM induces a pseudo-metric in feature space which we use to regularize feature drift in important directions and to update Gaussian prototypes used in a novel asymmetric cross entropy loss which effectively balances prototype rehearsal with data from new tasks. Experimental results on CIFAR-100, Tiny-ImageNet, ImageNet-Subset and ImageNet-1K demonstrate that Elastic Feature Consolidation is better able to learn new tasks by maintaining model plasticity and significantly outperform the state-of-the-art.[†]

## 1 INTRODUCTION

Deep neural networks achieve state-of-the-art performance on a broad range of visual recognition problems. However, the traditional supervised learning paradigm is limited in that it presumes all training data for all tasks is available in a single training session. The goal of Class-Incremental Learning (CIL) is to enable incremental integration of new classification tasks into already-trained models as they become available (Masana et al., 2022). Class-incremental learning entails balancing model *plasticity* (to allow learning of new tasks) against model *stability* (to avoid catastrophic forgetting of previous tasks) (McCloskey & Cohen, 1989). *Exemplar-based* approaches retain a small set of samples from previous tasks which are replayed to avoid forgetting, while *exemplar-free* methods retain no samples from previous tasks. This latter category is of particular interest when retaining exemplars is problematic due to storage or privacy requirements.

Exemplar-free Class Incremental Learning (EFCIL) must mitigate catastrophic forgetting without recourse to stored samples from previous tasks. Approaches can be loosely grouped into those based on weight regularization and those based on functional regularization. Elastic Weight Consolidation (EWC) is an elegant approach based on a Laplace approximation of the previous-task posterior (Kirkpatrick et al., 2017). When training on a new task, an approximate Fisher Information Matrix is used to regularize parameter drift in directions of significant importance to previous tasks. This mitigates forgetting while maintaining more plasticity for learning the new task. EWC is a second-order approach, and as such needs aggressive Fisher Information Matrix approximations (Liu et al., 2018).

---

[*]Corresponding author

[†]Code to reproduce experiments is available at `https://github.com/simomagi/elastic_feature_consolidation`

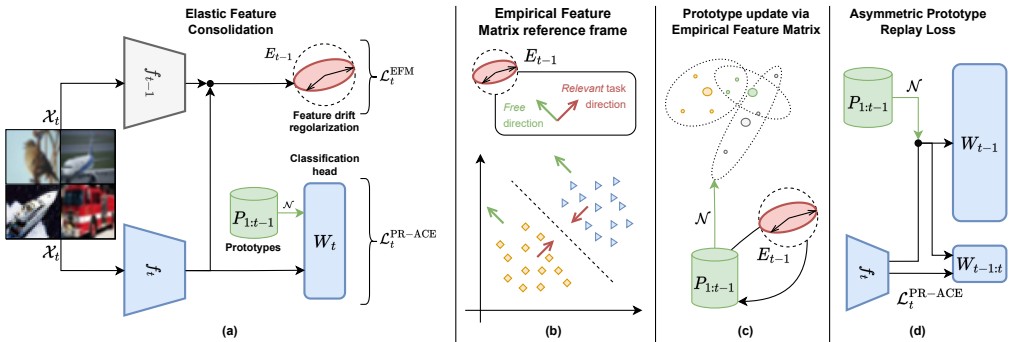

Figure 1: Elastic Feature Consolidation. (a) Architecture overview; (b) The Empirical Feature Matrix (EFM) measures how outputs vary with features and identifies important directions to mitigate forgetting (Section 3.3); (c) The EFM induces a pseudo-metric in feature space used to estimate prototype drift (Section 4.3); and (d) The Asymmetric Prototype Replay loss adapts previous task classifiers to the changing backbone by balancing new-task data and Gaussian prototypes (Section 4.2).

Functional regularization, especially through feature distillation, is a central component in recent state-of-the-art EFCIL approaches (Zhu et al., 2021b; 2022; Yu et al., 2020; Toldo & Ozay, 2022). Instead of regularizing weight drift, feature distillation regularizes drift in feature space to mitigate forgetting. Feature distillation is often combined with class prototypes, either learned (Toldo & Ozay, 2022) or based on class means (Zhu et al., 2021b), to perform pseudo-rehearsal of features from previous tasks which reduces the task-recency bias common in EFCIL (Masana et al., 2022). Prototypes – differently than exemplars – offer a privacy-preserving way to mitigate forgetting. Feature distillation and prototype rehearsal reduce feature drift across tasks, but at the cost of plasticity.

Existing EFCIL methods are predominantly evaluated in *Warm Start* scenarios in which the first task contains a larger number of classes than the rest, typically 50% or 40% of the entire dataset. For Warm Start scenarios* stability is more important than plasticity, since a good backbone can be already learned on the large first task. Current state-of-the-art approaches use strong backbone regularization (Zhu et al., 2021b) or indeed even *freeze* backbone after the large first task and focus on incrementally training the classifier (Petit et al., 2023).

In this paper we consider EFCIL in the more challenging *Cold Start* scenario in which the first task is insufficiently large to learn a high-quality backbone and methods must be plastic and adapt their backbone to new tasks. EFCIL with Cold Starts faces two main challenges: an alternative to feature distillation is required, since the backbone must adapt to new data, and an exemplar-free mechanism is needed to adapt previous-task classifiers to the changing backbone.

We propose a novel EFCIL approach, which we call Elastic Feature Consolidation (EFC), that regularizes changes in directions in *feature space* most relevant for previously-learned tasks and allows more plasticity in other directions. A main contribution of this work is the derivation of a pseudo-metric in feature space that is induced by a matrix we call the Empirical Feature Matrix (EFM). In contrast to the Fisher Information Matrix, the EFM can be easily stored and calculated since it does not depend on the number of model parameters, but only on feature space dimensionality. To address drift of the more plastic backbone, we additionally propose an Asymmetric Prototype Replay loss (PR-ACE) that balances between new-task data and Gaussian prototypes during EFCIL. Finally, we propose an improved method to update the class prototypes which exploits the already-computed EFM. A visual overview of EFC and its components is given in Figure 1.

## 2 RELATED WORK

We focus on **offline, class-incremental** learning problems. Class-incremental problems are distinguished from task-incremental problems in that **no task information is available at test time**. Offline incremental learning allows multiple passes on task datasets during training (Rebuffi et al., 2017; Wu et al., 2019; Masana et al., 2022), while online incremental learning considers continuous data

---

*We define Warm Start, differently than Ash & Adams (2020), as continual learning with a large initial task, and Cold Start, consistent with Hu et al. (2021), for continual learning beginning with a random network initialization.

streams in which each sample is accessed just once during training (Lopez-Paz & Ranzato, 2017; Mai et al., 2022; Soutif-Cormerais et al., 2023).

Exemplar-based approaches rely on a memory of previous class samples that are replayed while learning new tasks. Hou et al. (2019); Douillard et al. (2020); Kang et al. (2022) combine cosine normalization and distillation losses with exemplars to mitigate forgetting. Another category of exemplar-based approaches focuses on expanding sub-network structures across the incremental steps. Liu et al. (2021) adds two residual blocks to mask layers and balance stability and plasticity. Yan et al. (2021) train a single backbone for each incremental task, and more recently Zhou et al. (2022) share generalized residual blocks and extend specialized blocks for new tasks to improve performance and efficiency. The main drawbacks of exemplar-based approaches are the lack of privacy preservation and high computational and storage costs, particularly when a growing memory buffer is used.

Exemplar-free approaches are less common in class-incremental learning. Early works computed importance scores for all weights and used them to perform weight regularization (Kirkpatrick et al., 2017; Liu et al., 2018; Ritter et al., 2018). Other works like Li & Hoiem (2017); Jung et al. (2016) use functional regularization, such as knowledge or feature distillation, constraining network activations to match those of a previously-trained network. Most functional regularization approaches use feature distillation (Zhu et al., 2021a; Yu et al., 2020), or even freeze the feature extractor after the first task (Petit et al., 2023; Belouadah & Popescu, 2018; Panos et al., 2023), greatly reducing the plasticity of the network. For this reason, they consider Warm Start scenarios in which the first task is much larger that the others, so that reducing plasticity after the first task has less impact on the performance.

On top of this decrease in plasticity, only using functional regularization is not enough to learn new boundaries between classes from previous tasks and classes from new ones. For this reason regularization is often combined with *prototype rehearsal* Zhu et al. (2021b; 2022); Smith et al. (2021). Prototypes are feature space statistics (typically class means) used to reinforce the boundaries of previous-task classes without the need to preserve exemplars. Zhu et al. (2021b;a) use prototype augmentation and self-supervised learning to learn transferable features for future tasks. Toldo & Ozay (2022) learns and updates prototype representations during incremental learning. Zhu et al. (2022) combine prototypes with a strategy to re-organize network structure to transfer invariant knowledge across the tasks. Petit et al. (2023) fixes the feature extractor after the training the large first task and generates pseudo-samples to train a linear model discriminating all seen classes.

We propose an EFCIL approach that uses functional regularization and prototype rehearsal, but in contrast with most state-of-the-art methods, we also report results in *Cold Start* incremental learning scenarios that do not start with a large first task.

## 3 EFCIL REGULARIZATION VIA THE EMPIRICAL FEATURE MATRIX

In this section we introduce the general EFCIL framework and then derive a novel pseudo-metric in feature space, induced by a positive semi-definite matrix we call the *Empirical Feature Matrix* (EFM), used as a regularizer to control feature drift during class-incremental learning (see Figure 1b).

### 3.1 EXEMPLAR-FREE CLASS-INCREMENTAL LEARNING (EFCIL)

During class-incremental learning a model $\mathcal{M}$ is sequentially trained on $K$ tasks, each characterized by a disjoint set of classes $\{\mathcal{C}_t\}_{t=1}^{K}$, where each $\mathcal{C}_t$ is the set of labels associated to the task $t$. We denote the incremental dataset as $\mathcal{D} = \{\mathcal{X}_t, \mathcal{Y}_t\}_{t=1}^{K}$, where $\mathcal{X}_t$ and $\mathcal{Y}_t$ are, respectively, the set of samples and the set of labels for task $t$. The incremental model $\mathcal{M}_t$ at task $t$ consists of a feature extraction backbone $f_t(\cdot; \theta_t)$ shared across all tasks and whose parameters $\theta_t$ are updated during training, and a classifier $W_t \in \mathbb{R}^{n \times \sum_{j=1}^{t} |\mathcal{C}_j|}$ which grows with each new task. The model output at task $t$ is the composition of feature extractor and classifier:

$$\mathcal{M}_t(x; \theta_t, W_t) \equiv p(y|x; \theta_t, W_t) = \text{softmax}(W_t^\top f_t(x; \theta_t)). \tag{1}$$

Directly training $\mathcal{M}_t$ (i.e. fine-tuning) with a cross-entropy loss at each task $t$ results in backbone drift because at task $t$ we see no examples from previous-task classes. This progressively invalidates the previous-task classifiers and results in forgetting. In addition to the cross-entropy loss, many state-of-the-art EFCIL approaches use a regularization loss $\mathcal{L}_t^{\text{reg}}$ to reduce backbone drift, and a prototype loss $\mathcal{L}_t^{\text{pr}}$ to adapt previous-task classifiers to new features.

### 3.2 WEIGHT REGULARIZATION AND FEATURE DISTILLATION

Kirkpatrick et al. (2017) showed that using $\ell_2$ regularization to constrain weight drift reduces forgetting but does not leave enough plasticity for learning of new tasks. Hence, they proposed Elastic Weight Consolidation (EWC), which relaxes the $\ell_2$ constraint with a quadratic constraint based on a diagonal approximation of the Empirical Fisher Information Matrix (E-FIM):

$$F_t = \mathbb{E}_{x \sim \mathcal{X}_t} \left[ \mathbb{E}_{y \sim p(y|x;\theta_t^*)} \left\{ \left( \frac{\partial \log p}{\partial \theta_t^*} \right) \left( \frac{\partial \log p}{\partial \theta_t^*} \right)^T \right\} \right], \tag{2}$$

where $\theta_t^*$ is the model trained after task $t$. The E-FIM induces a pseudo-metric in parameter space (Amari, 1998) encouraging parameters to stay in a low-error region for previous-task models:

$$\mathcal{L}_t^{\text{E-FIM}} = \lambda_{\text{E-FIM}} (\theta_t - \theta_{t-1}^*)^T F_{t-1} (\theta_t - \theta_{t-1}^*). \tag{3}$$

The main limitation of approaches of this type is need for approximations of the E-FIM (e.g. a diagonal assumption). This makes the computation of the E-FIM tractable, but in practice is unrealistic since it does not consider off-diagonal entries that represent influences of interactions between weights on the log-likelihood. To overcome these limitations, more recent EFCIL approaches (Zhu et al., 2021b; 2022) rely on feature distillation (FD), which scale better in the number of parameters:

$$\mathcal{L}_t^{\text{FD}} = \sum_{x \in \mathcal{X}_t} ||f_t(x) - f_{t-1}(x)||_2. \tag{4}$$

The isotropic regularizer using the $\ell_2$ distance is too harsh a constraint for learning new tasks. We propose to use a pseudo-metric in *feature space*, induced by a matrix which we call the *Empirical Feature Matrix* (EFM), constraining directions in *feature space* most important for previous tasks, while allowing more plasticity in other directions when learning new tasks.

### 3.3 THE EMPIRICAL FEATURE MATRIX

Our goal is to regularize feature drift, but we do not want to do so isotropically as in feature distillation. Instead, we draw inspiration from how the E-FIM identifies important directions in parameter space and take a similar approach in feature space.

**The local and empirical feature matrices.** Let $f_t(x)$ be the feature vector extracted from input $x \in \mathcal{X}_t$ from task $t$ and $p(y) = p(y|f_t(x); W_t)$ the discrete probability distribution over all the classes seen so far. We define the *local feature matrix* as:

$$E_{f_t(x)} = \mathbb{E}_{y \sim p(y)} \left[ \left( \frac{\partial \log p(y)}{\partial f_t(x)} \right) \left( \frac{\partial \log p(y)}{\partial f_t(x)} \right)^\top \right]. \tag{5}$$

The Empirical Feature Matrix (EFM) associated with task $t$ is obtained by taking the expected value of Eq. 5 over the entire dataset at task $t$:

$$E_t = \mathbb{E}_{x \sim \mathcal{X}_t} [E_{f_t(x)}]. \tag{6}$$

Both $E_{f_t(x)}$ and $E_t$ are symmetric, as they are weighted sums of symmetric matrices, and positive semi-definite. Naively computing $E_{f_t(x)}$ requires a complete forward pass and a backward pass up to the feature embedding, however we exploit an analytic formulation that avoids computing gradients:

$$E_{f_t(x)} = \mathbb{E}_{y \sim p(y)} [W_t (I_m - P)_y (W_t (I_m - P)_y)^\top], \tag{7}$$

where $m = \sum_{j=1}^t |\mathcal{C}_j|$, $I_m$ the identity matrix of dimension $m \times m$, and $P$ is a matrix built from the row of softmax outputs associated with $f_t(x)$ replicated $m$ times (see Appendix B for a derivation).

To gain additional insight into what this matrix measures, we highlight some (information) geometrical aspects of $E_t$. The Empirical Fisher Information Matrix provides information about the information geometry of parameter space. In particular, it can be obtained as the second derivative of the KL-divergence for small weight perturbations (Martens, 2020; Grementieri & Fioresi, 2022). Similarly, the Empirical Feature Matrix provides information about the information geometry of *feature* space. Since $E_{f_t(x)}$ is positive semi-definite, we can interpret it as a pseudo-metric in feature space and use it to understand which perturbations of features most affect the predicted probability:

$$\text{KL}(p(y \mid f_t(x; \theta_t) + \delta; W_t) \;||\; p(y \mid f_t(x; \theta_t); W_t)) = \delta^T E_{f_t(x)} \delta + \mathcal{O}(||\delta||^3). \tag{8}$$

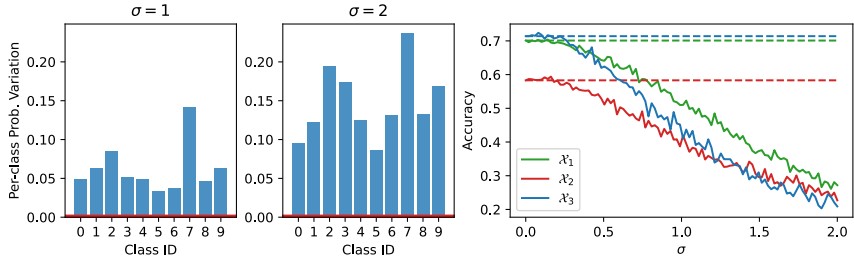

Figure 2: The regularizing effects of $E_t$ on Cold Start CIFAR-100 - 10 step (see Section 5.1 for more details on the dataset settings). **Left**: Perturbing features in principal directions of $E_1$ results in significant changes in classifier outputs (in blue), while perturbations in non-principal directions leave the outputs unchanged (in red). **Right**: If we continue incremental learning up through task 3 and perturb features from all three tasks in the principal (solid lines) and non-principal (dashed lines) directions of $E_3$, we see that $E_3$ captures *all* important directions in feature space up through task 3.

**The Empirical Feature Matrix loss.** We now have all the ingredients for our regularization loss:

$$\mathcal{L}_t^{\text{EFM}} = \mathbb{E}_{x \in \mathcal{X}_t} \left[ (f_t(x) - f_{t\text{-}1}(x))^T (\lambda_{\text{EFM}} E_{t\text{-}1} + \eta I)(f_t(x) - f_{t\text{-}1}(x)) \right], \tag{9}$$

where $f_t(x)$ and $f_{t-1}(x) \in \mathbb{R}^n$ are the features of sample $x$ extracted from the current model $\mathcal{M}_t$ and the model $\mathcal{M}_{t-1}$ trained on the previous task, respectively. In Eq. 9 we employ a damping term $\eta \in \mathbb{R}$ to constrain features to stay in a region where the quadratic second-order approximation of the KL-divergence of the log-likelihood in Eq. 8 remains valid (Martens & Sutskever, 2012). However, to ensure that our regularizer does not degenerate into feature distillation, we must ensure that $\lambda_{\text{EFM}} \mu_i > \eta$, where $\mu_i$ are the non-zero eigenvalues of the EFM. Additional analysis of these constraints and the spectrum of the EFM are given in Appendix B.2. In the next section we give empirical evidence of the effect of this elastic regularization.

**Regularizing effect of the EFM.** By definition, the directions of the eigenvectors of the EFM associated with strictly positive eigenvalues are those in which perturbations of the features have the most significant impact on predictions. To empirically verify this behavior, we perturb features in these *principal* directions and measure the resulting variation in probability outputs. Let $f_t = f_t(x) \in \mathbb{R}^n$ denote feature vector extracted from $x$ and $E_t$ the EFM computed after training on task $t$. We define the perturbation vector $\varepsilon_{1:k} \in \mathbb{R}^n$ such that each of the first $k$ entries is sampled from a Gaussian distribution $\mathcal{N}(0, \sigma)$, where $k$ is the number of strictly positive eigenvalues of the spectral decomposition of $E_t$. The remaining $n$-$k$ entries are set to zero. Then, letting $U$ be the matrix whose columns are the eigenvectors of $E_t$, we compute the perturbed features vector $\tilde{f}_t$ as follows:

$$\tilde{f}_t = f + U_t^\top \varepsilon_{1:k}. \tag{10}$$

In Figure 2 (left) we show that perturbing the feature space along the principal directions in this way, after training on the first task, results in a substantial variation in the probabilities for each class. Conversely, applying the complementary perturbation $\varepsilon_{k:n}$, i.e., setting zero on the first $k$ directions and applying Gaussian noise on the remaining $n$-$k$ directions, has no impact on the output. In Figure 2 (right) we see that, after training three tasks, perturbing in the principal directions of $E_3$ degrades performance across all tasks, while perturbations in non-principal directions continue to have no affect on the accuracy. This shows that $E_3$ captures the important feature directions for all previous tasks and that regularizing drift in these directions should mitigate forgetting.

## 4 PROTOTYPE REHEARSAL FOR ELASTIC FEATURE CONSOLIDATION

In this section we first describe prototype rehearsal and then describe our proposed Asymmetric Prototype Rehearsal strategy (see Figure 1d). We also show how the Empirical Feature Matrix $E_t$ defined above can be used to guide classifier drift compensation (see Figure 1c).

### 4.1 PROTOTYPE REHEARSAL FOR EXEMPLAR-FREE CLASS-INCREMENTAL LEARNING

EFCIL suffers from the fact that the final classifier is never jointly trained on the all seen classes. At each task $t$, the classifier $W_{t-1}$ is extended to include $|\mathcal{C}_t|$ new outputs. Thus, if classifier $W_t$ is only trained on samples from the current classes $\mathcal{C}_t$, it becomes incapable of discriminating classes

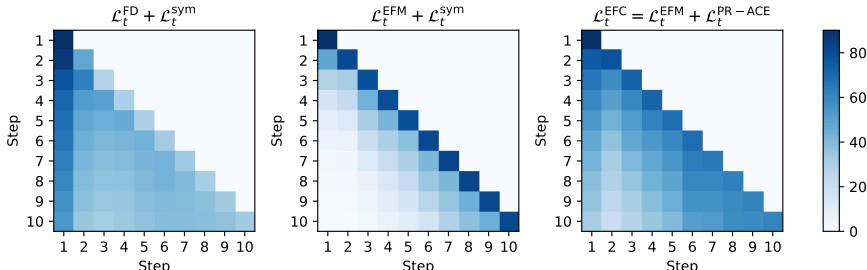

Figure 3: Accuracy after each incremental step on the Cold Start CIFAR-100 10-step scenario. Feature distillation with symmetric prototype loss (left) reduces forgetting at the cost of plasticity and new tasks are not learned. EFM regularization with symmetric loss (middle) increases plasticity at the cost of stability and previous tasks are forgotten. EFM regularization with asymmetric PR-ACE loss (right), balances current task data with prototypes and achieves better plasticity/stability trade-off.

from previous tasks, leading to inter-task confusion. Additionally, the classifier is biased towards the newest classes, resulting in task-recency bias (Masana et al., 2022).

A common *exemplar-free* solution is to replay only *prototypes* of previous classes (Zhu et al., 2022; 2021b;a; Petit et al., 2023; Toldo & Ozay, 2022). At the end of task $t-1$, the prototypes $p_{t-1}^c$ are computed as the class means of deep features and the set of prototypes $\mathcal{P}_{1:t-1}$ is stored.

During training of the next task the prototypes are perturbed and fed to the classification heads. Let $(\tilde{p}, y_{\tilde{p}}) \sim \mathcal{P}_{1:t-1}$ be a batch of perturbed prototypes and $(x_t, y_t) \sim (\mathcal{X}_t, \mathcal{Y}_t)$ a batch of current task data. The final classification loss, which we call the *symmetric* loss to distinguish it from the *asymmetric* loss we propose below, consists of two cross entropy losses evaluated on the all classification heads:

$$\mathcal{L}_t^{\text{sym}} = \mathcal{L}_t^{\text{ce}}(x_t, y_t)\big|_{\mathcal{C}_{1:t}} + \lambda_{\text{pr}} \, \mathcal{L}_t^{\text{ce}}(\tilde{p}, y_{\tilde{p}})\big|_{\mathcal{C}_{1:t}}, \tag{11}$$

where $\mathcal{L}_t^{\text{ce}}(x_t, y_t)\big|_{\mathcal{C}_{1:t}}$ represents the cross-entropy loss computed over all encountered classes $C_{1:t}$. In our work, we use Gaussian prototypes to enhance the feature class means, as suggested by Zhu et al. (2021a). To be more specific, we create augmented prototypes by sampling from a Gaussian distribution $\mathcal{N}(p_{t-1}^c, \Sigma^c)$. Here, $\Sigma^c$ represents the class covariance matrix of feature representations.

### 4.2 Asymmetric Prototype Rehearsal

$\mathcal{L}_t^{\text{sym}}$ from Eq. 11 above is able to balance old task classifiers with new ones only when feature space drift is small, for instance when using feature distillation (see Figure 3 (left)). But with the greater plasticity introduced by the EFM, the symmetric loss fails to effectively adapt old task classifiers to the changing backbone (see Figure 3 (middle)). To address this, we propose an *Asymmetric Prototype Replay loss* (PR-ACE) to balance current task data and prototypes:

$$\mathcal{L}_t^{\text{PR-ACE}} = \mathcal{L}_t^{\text{ce}}(x_t, y_t)\big|_{\mathcal{C}_t} + \mathcal{L}_t^{\text{ce}}((\tilde{p}, y_{\tilde{p}}) \cup (\hat{x}_t, \hat{y}_t))\big|_{\mathcal{C}_{1:t}}, \tag{12}$$

where $(x_t, y_t), (\hat{x}_t, \hat{y}_t) \sim (\mathcal{X}_t, \mathcal{Y}_t)$ and $\tilde{p} \sim \mathcal{P}_{1:t-1}$ with $y_{\tilde{p}}$ the associated label. $\mathcal{L}_t^{\text{ce}}(x, y)\big|_{\mathcal{C}^t}$ is the cross entropy loss restricted to classes in $\mathcal{C}^t$. The first term relies on the current task data, processed through the backbone and the current task classification head. It aligns the initially random head with the feature representation learned during previous tasks and aids learning of task-specific features. The second term calibrates the classification heads. It combines a batch of current task data with a batch of prototypes. Prototypes and current task data are uniformly sampled from all classes and used to train all task classifiers; consequently a larger proportion of prototypes is used compared to the current task data. In Figure 3 (right) we see that the addition of the PR-ACE loss achieves just this balance between stability and plasticity.

PR-ACE takes inspiration from the asymmetric loss proposed by Caccia et al. (2021) for *online, exemplar-based* class incremental learning. In their work, the goal was to learn inter-task features with a loss term based on both exemplars and current task data passed through the current backbone. In contrast, for prototype rehearsal only *current task data* may be passed through the backbone.

### 4.3 Prototype Drift Compensation via EFM

Fixed prototypes can become inaccurate over time due to drift from previous class representations. Yu et al. (2020) showed that the drift in *embedding networks* can be estimated using current task data.

They suggest estimating the drift of class means from previous-tasks by looking at feature drift in closely related classes of current task. After each task, they update the prior normalized class means via a weighted average of the current task feature representation drift. These updated class means are then used for nearest class-mean classification at test time.

Our aim is to update prototypes $p_{t-1}^c$ used in for rehearsal in PR-ACE (Eq. 12) across the entire training phase. As proposed by SDC, after each task we update the prototypes as follows:

$$\hat{p}_{t-1}^c = p_{t-1}^c + \frac{\sum_{x_i \sim \mathcal{X}_t} w_i \delta_i^{t-1}}{\sum_{x_i \sim \mathcal{X}_t} w_i}, \quad \delta_i^{t-1} = f_t(x_i) - f_{t-1}(x_i), \tag{13}$$

where $\delta_i^{t-1}$ is the feature drift of samples $x_i$ between task $t-1$ and $t$.

We define the weights $w_i$ using the EFM (Figure 1c). In Section 3.3 we showed that the EFM induces a pseudo-metric in feature space, providing a second-order approximation of the KL-divergence due to feature drift. We extend the idea of feature drift estimation to softmax-based classifiers by weighting the overall drift of our class prototypes. Writing $p^c = p_{t-1}^c$ and $f_{t-1}^i = f_{t-1}(x_i)$ to simplify notation, our weights are defined as:

$$w_i = \exp\left(-\frac{(f_{t-1}^i - p^c)E_{t-1}(f_{t-1}^i - p^c)^\top}{2\sigma^2}\right) \approx \exp\left(-\frac{\mathrm{KL}(p(y|f_{t-1}^i; W_{t-1}) \| p(y|p^c; W_{t-1}))}{2\sigma^2}\right), \tag{14}$$

where $E_{t-1}$ is the EFM after training task $t-1$. Eq. 14 assigns higher weights to the samples more closely aligned to prototypes in terms of probability distribution. Specifically, higher weights are assigned to samples whose softmax prediction matches that of the prototypes, indicating a strong similarity for the classifier. See Appendix C for more details.

### 4.4 Elastic Feature Consolidation with Asymmetric Prototype Rehearsal

Figure 1 summarizes our approach, which is a combination of the EFM and PR-ACE losses:

$$\mathcal{L}_t^{\mathrm{EFC}} = \mathcal{L}_t^{\mathrm{EFM}} + \mathcal{L}_t^{\mathrm{PR\text{-}ACE}}, \tag{15}$$

where $\mathcal{L}_t^{\mathrm{PR\text{-}ACE}}$ is the asymmetric cross entropy loss (Eq. 12) and $\mathcal{L}_t^{\mathrm{EFM}}$ is the Empirical Feature Matrix loss from (Eq. 9). After each task, our prototypes are updated using (Eq. 13) equipped with the EFM (Eq. 14) and new prototypes with the corresponding class covariances are computed. Finally, we compute the EFM $E_t$ using the current task data for use when learning subsequent incremental tasks. In Appendix D we provide the pseudocode of the overall training procedure.

Note that it is essential to combine $\mathcal{L}^{\mathrm{EFC}}$ with $\mathcal{L}_t^{\mathrm{PR\text{-}ACE}}$. $E_{t-1}$ *selectively* inhibits drift in important feature directions, in contrast to Feature Distillation which inhibits drift in *all* directions (Figure 3 (left)). The resulting plasticity increases the risk of prototype drift, which in turn adapts previous-task classifiers to drifted prototypes (Figure 3 (middle)). Our use of $\mathcal{L}_t^{\mathrm{PR\text{-}ACE}}$ to balance prototypes and new task samples, and our prototype drift compensation, counters this effect (Figure 3 (right)).

## 5 Experimental results

In this section we compare Elastic Feature Consolidation with the state-of-the-art in EFCIL.

### 5.1 Datasets, Metrics, and Hyperparameters

We consider three standard datasets: **CIFAR-100** (Krizhevsky et al., 2009),**Tiny-ImageNet** (Wu et al., 2017) and **ImageNet-Subset** (Deng et al., 2009). Each is evaluated in two settings. The first is the **Warm Start (WS)** scenario commonly considered in EFCIL Zhu et al. (2021b; 2022); Petit et al. (2023); Toldo & Ozay (2022) which uses a larger first task consisting of 50 classes for CIFAR-100 and ImageNet-Subset in the 10-step scenario, and 40 classes in the 20-step scenario. For Tiny-ImageNet, both the 10-step and 20-step scenarios use a large first task consisting of 100 classes. Regardless of 10-step or 20-step, the remaining classes after the first task are uniformly distributed among the subsequent tasks. The second setting, referred to as **Cold Start (CS)**, uniformly distributes all classes among all tasks. We are mostly interested in the Cold Start scenario since it requires backbone plasticity and is very challenging for EFCIL. In Appendix G we report the performance on full **ImageNet-1K** dataset. See Appendix E for more dataset details. We use as metrics the **per-step incremental accuracy** $A_{\mathrm{step}}^K$ and **average incremental accuracy** $A_{\mathrm{inc}}^K$:

$$A_{\mathrm{step}}^K = \frac{\sum_{i=1}^K |\mathcal{C}_i| a_i^K}{\sum_{i=1}^K |\mathcal{C}_i|}, \quad A_{\mathrm{inc}}^K = \frac{1}{K} \sum_{i=1}^K A_{\mathrm{step}}^i, \tag{16}$$

| | Method | Warm Start $A_{\text{step}}^K$ 10 Step | 20 Step | $A_{\text{inc}}^K$ 10 Step | 20 Step | Cold Start $A_{\text{step}}^K$ 10 Step | 20 Step | $A_{\text{inc}}^K$ 10 Step | 20 Step |
|---|---|---|---|---|---|---|---|---|---|
| CIFAR-100 | EWC | $21.08 \pm 1.09$ | $13.53 \pm 1.11$ | $41.00 \pm 1.11$ | $31.79 \pm 2.77$ | $31.17 \pm 2.94$ | $17.37 \pm 2.43$ | $49.14 \pm 1.28$ | $31.02 \pm 1.15$ |
| | LwF | $20.73 \pm 1.47$ | $11.78 \pm 0.60$ | $41.95 \pm 1.30$ | $28.93 \pm 1.62$ | $32.80 \pm 3.08$ | $17.44 \pm 0.73$ | $\underline{53.91} \pm 1.67$ | $38.39 \pm 1.05$ |
| | PASS | $53.42 \pm 0.48$ | $47.51 \pm 0.37$ | $63.42 \pm 0.69$ | $59.55 \pm 0.97$ | $30.45 \pm 1.01$ | $17.44 \pm 0.69$ | $47.86 \pm 1.93$ | $32.86 \pm 1.03$ |
| | Fusion | $56.86$ | $51.75$ | $65.10$ | $61.60$ | – | – | – | – |
| | FeTrIL | $56.79 \pm 0.40$ | $52.61 \pm 0.81$ | $65.03 \pm 0.66$ | $62.50 \pm 1.03$ | $\underline{34.94} \pm 0.46$ | $\underline{23.28} \pm 1.24$ | $51.20 \pm 1.13$ | $\underline{38.48} \pm 1.07$ |
| | SSRE | $\underline{57.48} \pm 0.55$ | $\underline{52.98} \pm 0.63$ | $\underline{65.78} \pm 0.59$ | $\underline{63.11} \pm 0.84$ | $30.40 \pm 0.74$ | $17.52 \pm 0.80$ | $47.26 \pm 1.91$ | $32.45 \pm 1.07$ |
| | **EFC** | $\mathbf{60.87} \pm 0.39$ | $\mathbf{55.78} \pm 0.42$ | $\mathbf{68.23} \pm 0.68$ | $\mathbf{65.90} \pm 0.97$ | $\mathbf{43.62} \pm 0.70$ | $\mathbf{32.15} \pm 1.33$ | $\mathbf{58.58} \pm 0.91$ | $\mathbf{47.36} \pm 1.37$ |
| Tiny-ImageNet | EWC | $6.73 \pm 0.44$ | $5.96 \pm 1.17$ | $18.48 \pm 0.60$ | $13.74 \pm 0.52$ | $8.00 \pm 0.27$ | $5.16 \pm 0.54$ | $24.01 \pm 0.51$ | $15.70 \pm 0.35$ |
| | LwF | $24.00 \pm 1.44$ | $7.58 \pm 0.37$ | $43.15 \pm 1.02$ | $22.89 \pm 0.64$ | $26.09 \pm 1.29$ | $15.02 \pm 0.67$ | $45.14 \pm 0.88$ | $32.94 \pm 0.54$ |
| | PASS | $41.67 \pm 0.64$ | $35.01 \pm 0.39$ | $51.18 \pm 0.31$ | $46.65 \pm 0.47$ | $24.11 \pm 0.48$ | $18.73 \pm 1.43$ | $39.25 \pm 0.90$ | $32.01 \pm 1.68$ |
| | Fusion | $\underline{46.92}$ | $44.61$ | – | – | – | – | – | – |
| | FeTrIL | $45.71 \pm 0.39$ | $44.63 \pm 0.49$ | $\underline{53.95} \pm 0.42$ | $\underline{52.96} \pm 0.45$ | $\underline{30.97} \pm 0.90$ | $\underline{25.70} \pm 0.61$ | $\underline{45.60} \pm 1.67$ | $\underline{39.54} \pm 1.19$ |
| | SSRE | $44.66 \pm 0.45$ | $\underline{44.68} \pm 0.36$ | $53.27 \pm 0.43$ | $52.94 \pm 0.42$ | $22.93 \pm 0.95$ | $17.34 \pm 1.06$ | $38.82 \pm 1.99$ | $30.62 \pm 1.96$ |
| | **EFC** | $\mathbf{50.40} \pm 0.25$ | $\mathbf{48.68} \pm 0.65$ | $\mathbf{57.52} \pm 0.43$ | $\mathbf{56.52} \pm 0.53$ | $\mathbf{34.10} \pm 0.77$ | $\mathbf{28.69} \pm 0.40$ | $\mathbf{47.95} \pm 0.61$ | $\mathbf{42.07} \pm 0.96$ |
| ImageNet-Subset | EWC | $16.19 \pm 2.48$ | $10.66 \pm 1.74$ | $23.58 \pm 2.01$ | $18.05 \pm 1.10$ | $24.59 \pm 4.13$ | $12.78 \pm 1.95$ | $39.40 \pm 3.05$ | $26.95 \pm 1.02$ |
| | LwF | $21.89 \pm 0.52$ | $13.24 \pm 1.61$ | $37.15 \pm 2.47$ | $25.96 \pm 0.95$ | $\underline{37.71} \pm 2.53$ | $18.64 \pm 1.67$ | $\underline{56.41} \pm 1.03$ | $40.23 \pm 0.43$ |
| | PASS | $52.04 \pm 1.06$ | $44.03 \pm 2.19$ | $65.14 \pm 0.36$ | $58.88 \pm 2.15$ | $26.40 \pm 1.33$ | $14.38 \pm 1.22$ | $45.74 \pm 0.18$ | $31.65 \pm 0.42$ |
| | Fusion | $60.20$ | $51.60$ | $70.00$ | $63.70$ | – | – | – | – |
| | FeTrIL | $\underline{63.56} \pm 0.59$ | $\underline{57.62} \pm 1.13$ | $\underline{71.87} \pm 1.46$ | $\underline{68.01} \pm 1.60$ | $36.17 \pm 1.18$ | $\underline{26.63} \pm 1.45$ | $52.63 \pm 0.56$ | $\underline{42.43} \pm 2.05$ |
| | SSRE | $61.84 \pm 0.93$ | $55.19 \pm 0.97$ | $70.68 \pm 1.37$ | $66.73 \pm 1.61$ | $25.42 \pm 1.17$ | $16.25 \pm 1.05$ | $43.76 \pm 1.07$ | $31.15 \pm 1.53$ |
| | **EFC** | $\mathbf{68.85} \pm 0.58$ | $\mathbf{62.17} \pm 0.69$ | $\mathbf{75.40} \pm 0.92$ | $\mathbf{71.63} \pm 1.13$ | $\mathbf{47.38} \pm 1.43$ | $\mathbf{35.75} \pm 1.74$ | $\mathbf{59.94} \pm 1.38$ | $\mathbf{49.92} \pm 2.05$ |

Table 1: Comparison with the state-of-the-art on CIFAR-100, Tiny-ImageNet, and ImageNet-Subset. Fusion results are those reported in (Toldo & Ozay, 2022) as no code is provided to reproduce them.

where $a_i^K$ represents the accuracy of task $i$ after training task $K$.

**Hyperparameter settings.** We use the standard ResNet-18 (He et al., 2015) trained from scratch for all experiments. We train the first task of each state-of-the-art method using the same optimizer, number of epochs and data augmentation. In particular, we train all the first task using self-rotation as performed by Zhu et al. (2021b); Toldo & Ozay (2022) to align all the performance. In Appendix E we provide optimization settings for the first tasks.

For the incremental steps of EFC we used Adam with weight decay of $2e^{-4}$ and fixed learning rate of $1e^{-4}$ for Tiny-ImageNet and CIFAR-100, while for ImageNet-Subset we use a learning rate of $1e^{-5}$ for the backbone and $1e^{-4}$ for the heads. We fixed the total number of epochs to 100 and use a batch size of 64. We set $\lambda_{\text{EFM}} = 10$ and $\eta = 0.1$ in Eq. 9 for all the experiments. We ran all approaches using five random seeds and shuffling the classes in order to reduce the bias induced by the choice of class ordering (Rebuffi et al., 2017; Masana et al., 2022). In Appendix E we provide the optimization settings for each state-of-the art method we evaluated.

## 5.2 COMPARISON WITH THE STATE-OF-THE-ART

In Table 1 we compare EFC with baselines and state-of-the-art EFCIL approaches. Specifically we consider EWC (Kirkpatrick et al., 2017), LwF (Li & Hoiem, 2017), PASS (Zhu et al., 2021b), Fusion (Toldo & Ozay, 2022), FeTrIL (Petit et al., 2023) and SSRE (Zhu et al., 2022). Our evaluation considers two key scenarios: Warm Start, commonly found in the EFCIL literature, and the more difficult Cold Start scenario. EFC significantly outperforms the previous state-of-the-art in both metrics across all scenarios of the considered datasets.

Notably, on the ImageNet-Subset Warm Start scenario EFC exhibits a substantial improvement of about 5% over FeTrIL in both per-step and average incremental accuracy. For SSRE, our incremental accuracy results are significantly higher compared to the results reported in the original paper. This is attributable to the self-rotation we used as an initial task to align all results with PASS and Fusion (which both use self-rotation). This alignment is crucial when dealing with Warm Start scenarios, as performance on the large first task heavily biases the metrics (Eq. 16) as discussed by Zhou et al. (2023); Castro et al. (2018). In Appendix J and K we provide per-step performance plots in all the analyzed scenarios, showing that all the evaluated methods begin from the same starting point.

In Cold Start scenarios we see that methods relying on feature distillation, such as FeTrIL, SSRE and PASS, experience a significant decrease in accuracy compared to EFC. This drop in accuracy can be attributed to the fact that the first task does not provide a sufficiently strong starting point for the entire class-incremental process. Moreover, most of these approaches exhibit weaker performance even when compared to LwF, which does not use prototypes to balance the classifiers. On the contrary, the plasticity offered by the EFM allows Elastic Feature Consolidation to achieve excellent performance

| | Warm Start | | | | Cold Start | | | |
|---|---|---|---|---|---|---|---|---|
| | $A^K_{\text{step}}$ | | $A^K_{\text{inc}}$ | | $A^K_{\text{step}}$ | | $A^K_{\text{inc}}$ | |
| **Variant** | **10 Step** | **20 Step** | **10 Step** | **20 Step** | **10 Step** | **20 Step** | **10 Step** | **20 Step** |
| Each Task | 49.00 ±0.48 | 47.26 ±0.18 | 56.31 ±0.51 | 55.06 ±0.39 | **34.81** ±0.56 | **29.39** ±0.52 | 47.84 ±0.39 | **42.53** ±0.17 |
| Last Task | 48.00 ±0.43 | 45.64 ±0.51 | 55.85 ±0.54 | 54.12 ±0.37 | 34.36 ±0.14 | 28.57 ±0.32 | 47.50 ±0.24 | 42.11 ±0.11 |
| **EFC** | **50.40** ±0.25 | **48.68** ±0.65 | **57.52** ± 0.43 | **56.52** ±0.53 | 34.10 ±0.77 | 28.69 ±0.40 | **47.95** ±0.61 | 42.07 ±0.96 |

Table 2: Mitigating storage costs (on Tiny-ImageNet). **Each Task** stores a single covariance per task, while **Last Task** stores a single covariance from the most recent task (see Appendix I, Table A5).

even in this setting, surpassing all other approaches by a significant margin. In Appendix G we provide results comparing EFC with FeTrIL on ImageNet-1K.

## 5.3 ABLATION STUDY AND STORAGE COSTS

In this section we report the ablation study and the storage costs of EFC on CIFAR-100. A more detailed analysis on other dataset is reported in Appendices H and I.

**Ablation on PR-ACE and Feature Distillation.** We evaluated the performance of our regularizer in two settings: when combined with the proposed PR-ACE Eq. 12 and when combined with the standard *symmetric loss* Eq. 11. Figure 4a clearly demonstrates that the symmetric loss works well in conjunction with feature distillation. However, when used alongside the EFM, it fails to effectively control the adaptation of the old task classifier, resulting in an overall drop in performance. Additionally, it is evident from the results that feature distillation essentially degenerates into a method similar to FeTrIL, which freezes the backbone after the first task. These findings suggest that using feature distillation as a regularizer is equivalent to freezing the backbone after the first task and employing a suitable prototype rehearsal approach to balance the task classifier, pseudo-samples for FeTrIL, and Gaussian prototypes in our specific scenario.

**Ablation on prototype update.** In Figure 4b we evaluate the effect of prototype updates (Eq. 13 and Eq. 14). These results indicate that prototype updates enhance performance in both the Warm and the Cold Start experiments. The performance boost is more pronounced in the Cold Start scenario, where we achieve 4 and 5 points of improvement for the 10 and 20 steps respectively.

**Storage costs**. Storing class covariance matrices results in a linear increase in storage as the number of classes grows. This storage cost is easily mitigated, however, without sacrificing state-of-the-art accuracy. In Table 2 we consider the use of a task-specific covariance matrix which comes with a linear memory increase in the number of *tasks*, and using a single covariance from the most recent task for prototype generation which results in *constant* memory overhead. In Appendix I we provide results an analysis of using low-rank approximations of all covariance matrices which does not sacrifice EFCIL performance.

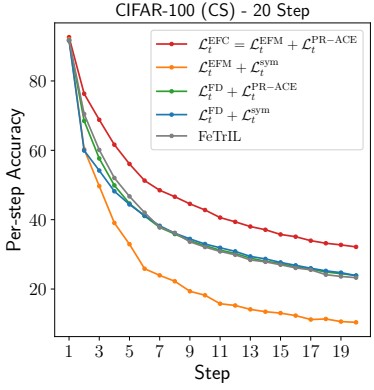

(a) Ablation on regularization and prototype losses.

| Step | Update Proto (WS) | | Update Proto (CS) | |
|---|---|---|---|---|
| | ✗ | ✓ | ✗ | ✓ |
| 10 | 58.24 | **60.87** | 39.28 | **43.62** |
| 20 | 52.60 | **55.78** | 27.28 | **32.15** |

(b) Ablation on prototype update.

Figure 4: Ablations on (a) losses and (b) prototype update for CIFAR-100.

## 6 CONCLUSIONS AND LIMITATIONS

Cold Start EFCIL requires careful balance between model stability and plasticity. In this paper we proposed Elastic Feature Consolidation which regularizes feature drift in directions in feature space important to previous-task classifiers. We derive an Empirical Feature Matrix that induces a pseudo-metric used to control feature drift and improves plasticity with respect to feature distillation. We further use this Empirical Feature Matrix to constrain prototype updates in an Asymmetric Prototype Replay loss. Our results show that EFC significantly outperforms the state-of-the-art on both Cold Start and Warm Start EFCIL. EFC updates class means based on current class sample drift, but stored means could be distant from real class means, potentially leading to forgetting in long task streams. Estimating covariance drift is still an open question. EFC needs storage that grows linearly with classes number. This growth can be mitigated using approximations or proxies for class covariance matrices (see Appendix I).

## REPRODUCIBILITY STATEMENT

We ran all experiments five times with different seeds and report the average and standard deviations in order to guarantee that reported results are reproducible and not due to random fluctuations in performance. We report all hyperparameters used to reproduce our results in Section 5.1 and Appendix E.2 – for Elastic Feature Consolidation and all methods we compare to. Source code to reproduce our experimental results is public released.

## ACKNOWLEDGEMENTS

This work was supported by funding by the European Commission Horizon 2020 grant #951911 (AI4Media), project and TED2021-132513B-I00 and PID2022-143257NB-I00 funded by MCIN/AEI/ 10.13039/501100011033, by the Italian national Cluster Project CTN01_00034_23154 "Social Museum Smart Tourism", by the European Union project NextGenerationEU/PRTR, and by the Spanish Development Fund FEDER.

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

# Appendices

## A SCOPE AND SUMMARY OF NOTATION

In these appendices we provide additional information, experimental results, and analyses that complement the main paper. For convenience, we summarize here the notation used in the paper:

- $\mathcal{M}_t$: The incremental model at task $t$.
- $K$: The total number of tasks.
- $\mathcal{C}_t$: The set of classes associated with task $t$.
- $\mathcal{D}$: The incremental dataset containing samples $\mathcal{X}_t$ and labels $\mathcal{Y}_t$ for each task $t$.
- $f_t(\cdot; \theta_t)$: The feature vector computed by the backbone at task $t$ with parameters $\theta_t$.
- $n$: The dimensionality of the feature space.
- $W_t$: The classifier at task $t$ with dimensions $\mathbb{R}^{n \times \sum_{j=1}^{t} |\mathcal{C}_j|}$.
- $\lambda_{\text{EFM}}$: hyperparameter associated to the Empirical Feature Matrix, Eq. 9 in the paper.
- $\eta$: damping hyperparameter associated to the Identity Matrix, Eq. 9 in the paper.

In Appendix B, we derive an analytic form for computing the Empirical Feature Matrix and analyze how its rank evolves during incremental learning. In Appendix C, we provide additional analysis on using the EFM to update prototypes. In Appendix D, we present the training loop employed in our method. In Appendix E, we provide additional information about the dataset and the implementation of the approaches we tested. In Appendix F, we provide a comprehensive evaluation of our proposed method on a short task sequence. In Appendix G, we compare the results in both Warm and Cold Start scenarios of EFC against FeTrIL on ImageNet-1K. In Appendix H, we extend the ablation study presented in Section 5.3 on the main paper. In Appendix I we analyze the storage and computation costs of EFC and propose techniques to mitigate them. Finally, in Appendix J and Appendix K we give per-step plots of accuracy on all datasets and scenarios.

## B THE EMPIRICAL FEATURE MATRIX

### B.1 ANALYTIC FORMULATION FOR COMPUTING THE EMPIRICAL FEATURE MATRIX

In this section we derive an analytic formulation of the local features matrix defined in the main paper in Section 3.3. To simplify the notation, we recall its definition without explicitly specifying the task, as it is unnecessary for the derivation:

$$E_{f(x)} = \mathbb{E}_{y \sim p(y)} \left[ \left( \frac{\partial \log p(y)}{\partial f(x)} \right) \left( \frac{\partial \log p(y)}{\partial f(x)} \right)^{\top} \right]. \tag{17}$$

We begin by computing the Jacobian matrix with respect to the feature space of the output function $g : \mathbb{R}^m \to \mathbb{R}^m$ which is the log-likelihood of the model on logits $z$:

$$g(z) = \log(\text{softmax}(z)) = [\log(\sigma_1(z)), \ldots, \log(\sigma_m(z))]^{\top},$$

where

$$\sigma_i(z) = \frac{e^{z_i}}{\sum_{j=1}^{m} e^{z_j}}.$$

The partial derivatives of $g$ with respect to each input $z_i$ are:

$$\frac{\partial \log(\sigma_i(z))}{\partial z_j} = \begin{cases} 1 - \sigma_i(z) & \text{if } i = j, \\ -\sigma_j(z) & \text{otherwise.} \end{cases} \tag{18}$$

Thus, the Jacobian matrix of $g$ is:

$$J(z) = \begin{bmatrix} 1 - \sigma_1(z) & -\sigma_2(z) & \dots & -\sigma_m(z) \\ -\sigma_1(z) & 1 - \sigma_2(z) & & \vdots \\ \vdots & & \ddots & \\ -\sigma_1(z) & & \dots & 1 - \sigma_m(z) \end{bmatrix} \tag{19}$$

$$= I_m - \mathbb{1}_m \cdot \begin{bmatrix} \sigma_1(z) \\ \vdots \\ \sigma_m(z) \end{bmatrix}^\top = I_m - P, \tag{20}$$

where $I_m \in \mathbb{R}^{m \times m}$ is the identity matrix and $\mathbb{1}_m$ represent the vector with all entries equal to 1.

Recalling Eq. 17, we are interested in computing the derivatives of the log-probability vector with respect to the feature vector:

$$\frac{\partial \log p(y_1, \dots, y_m | f; W)}{\partial f} = \frac{\partial (Wf)}{\partial f} \frac{\partial (\log(\text{softmax}(z)))}{\partial z} \tag{21}$$

$$= W J(z), \tag{22}$$

where $z = Wf$ for $W$ the classifier weight matrix mapping features $f$ to logits $z$. Combining this with Eq. 20 we have:

$$\mathbb{E}_{y \sim p(y)} \left[ \left( \frac{\partial \log p(y)}{\partial f} \right) \left( \frac{\partial \log p(y)}{\partial f} \right)^\top \right] = \mathbb{E}_{y \sim p(y)} [W(I_m - P)_y (W(I_m - P)_y)^\top], \tag{23}$$

where $p(y) = p(y|f(x); W)$, $P$ is the matrix containing the probability vector associated with $f$ in each row, and $(I_m - P)_y$ is the column vector containing the $y_{th}$ row of the Jacobian matrix. Computing $E_t$ using Eq. 23 requires only a single forward pass of all data through the network, whereas a naive implementation requires an additional backward pass up to the feature embedding layer.

## B.2 Spectral analysis of the Empirical Feature Matrix

In this section we conduct a spectral analysis on our Empirical Feature Matrix (EFM). Since Elastic Feature Consolidation (EFC) uses $E_t$ to selectively regularize drift in feature space, it is natural to question *how* selective its regularization is. We can gain insight into this by analyzing how the rank of $E_t$ evolves with increasing incremental tasks.

Let us consider a model incrementally trained on CIFAR-100 using EFC method up to the sixth and final task and compute the spectrum of the $E_t$ for each $t \in [1, \dots, 6]$. We chose a shorter task sequence for simplicity, but the same empirical conclusions hold regardless of the number of tasks. Recalling that $E_t \in \mathbb{R}^{n \times n}$ is symmetric, thus there exist $U_t$ and $\Lambda_t \in \mathbb{R}^{n \times n}$ such that:

$$E_t = U_t^{-1} \Lambda_t U_t = U_t^\top \Lambda_t U_t, \tag{24}$$

where $U_t$ is an orthogonal matrix and $\Lambda_t = \text{diag}(\lambda_1, .., \lambda_n)$ is a diagonal matrix whose entries are the eigenvalues of $E_t$. Since $E_t$ is positive semi-definite, $\lambda_i \geq 0$ for each $i \in [1, \dots, n]$. For our convenience we can rearrange the matrices $\Lambda_t$ and $U_t$ of the decomposition such that the eigenvalues on the diagonal are ordered by the greatest to the lowest, with the first $k$ being positive. In Figure A1, we plot the spectrum of $E_t$ at each $t \in [1, \dots, 6]$. The plot shows that the number of classes on which the matrix is estimated corresponds to an elbow in the curve beyond which the spectrum of the matrix vanishes. This implies that the rank of the matrices is exactly equal to the number of observed classes. This empirical observation is in line with the results shown in Grementieri & Fioresi (2022) where the authors introduce the concept of a *local data matrix* similar to our local feature matrix, with the distinction that they take derivatives with respect to inputs instead of features. In the paper they prove, for a simple classification task, that the local data matrix has rank equal to the number of classes minus one, which is exactly what we observe in the evolution of $E_t$ during incremental learning.

Finally, we highlight that the results reported for EFC in the various tables of the paper were obtained using parameter values of $\lambda_{\text{EFM}} = 10$ and $\eta = 0.1$ in the regularization loss (Eq. 9). In Figure A1, the horizontal line indicates that the plasticity constraints $\lambda_{\text{EFM}} \mu_i > \eta$ (described in Section 3.3 of the paper) are satisfied for nearly every $\mu_i > 0$. This observation clearly demonstrates that our regularizer effectively constrains the features in a non-isotropic manner, distinguishing it from feature distillation.

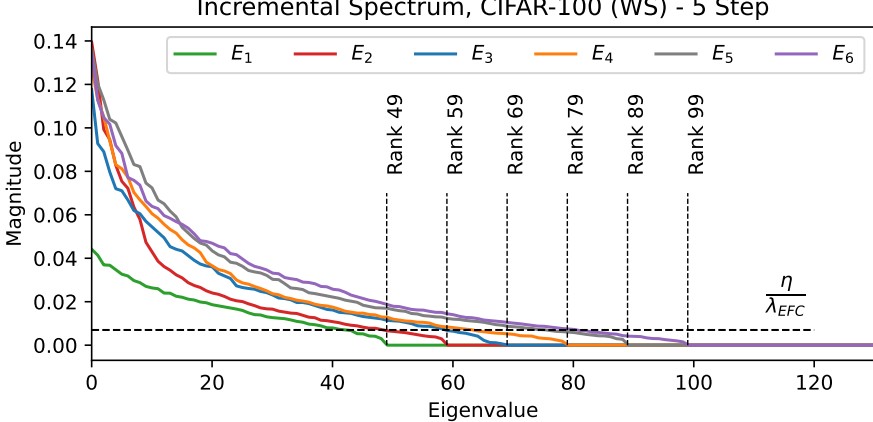

Figure A1: The spectrum of the Empirical Feature Matrix across incremental learning steps. For a better visualization of the spectrum in the analysis we considered a Warm Start 5-step scenario on CIFAR-100. The $x$-axis is truncated at the 120th eigenvalue.

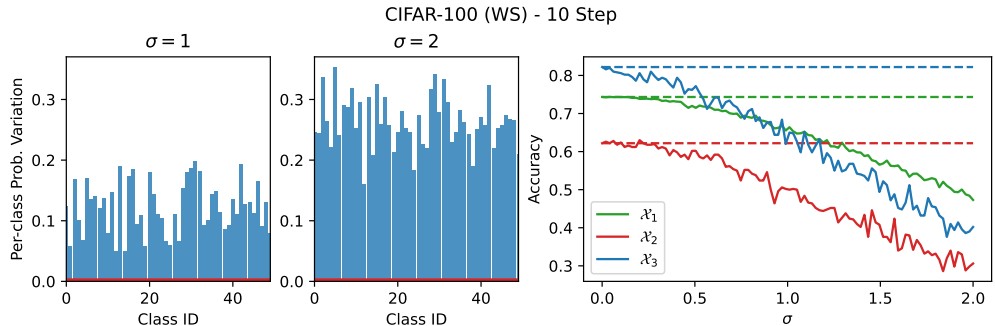

Figure A2: The regularizing effects of $E_t$ in the Warm Start Scenario. Principal and non-principal direction are represented by solid lines and dashed lines, respectively.

### B.3 REGULARIZING EFFECTS OF THE EMPIRICAL FEATURE MATRIX

In this section we extend the analysis provided in Section 3.3 and Figure 2 on CIFAR-100 (CS) - 10 steps. Our aim is to show that the regularizing effect of the Empirical Feature Matrix remains consistent regardless of the number of classes considered in the initial task, namely in the Warm Start scenario. Figure A2) shows that applying perturbation in *principal directions* of the EFM leads significant changes in classifier outputs, resulting in a degradation of performance. Conversely, when the perturbation is applied in the *non-principal* directions, neither the classifier probabilities nor the performance change. The results are in line with those observed in the same analysis but on the Cold Start scenario.

Finally, we demonstrate the robustness of this analysis with respect to the order of classes within the tasks in both scenarios. To achieve this, we consider five different random network initializations and class orderings. In Figure A3, we give the average accuracy drop when adding Gaussian perturbations along principal and non-principal directions (separately). The plot reveals that perturbations along non-principal directions have no impact on accuracy (indicated by three dashed, overlapped lines on the x-axis), while perturbations along the relevant direction consistently degrade the performance of the model on all the considered tasks.

### C PROTOTYPE UPDATE IMPLEMENTATION DETAILS

Here we provide some additional details about the prototype update using the Empirical Feature Matrix as described in the main paper in Section 4.3. We weight the drift to update the prototypes

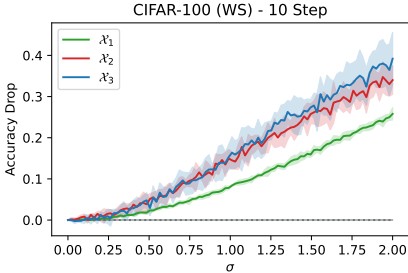 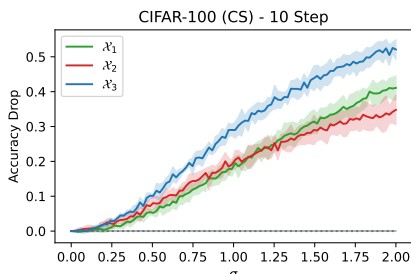

Figure A3: Average accuracy drop in the Warm Start (left) and in the Cold Start (right) scenarios after 3 incremental steps when perturbing along principal (solid lines) or non-principal (dashed lines) directions of $E_3$. The values are averaged over 5 seeds, and the shaded region around the lines represents the standard deviation.

using:

$$w_i = \exp\left(-\frac{(f_{t\text{-}1}^i - p^c)E_{t\text{-}1}(f_{t\text{-}1}^i - p^c)^\top}{2\sigma^2}\right) \approx \exp\left(-\frac{\mathrm{KL}(p(y|f_{t\text{-}1}^i; W_{t\text{-}1}) \,||\, p(y|p^c; W_{t\text{-}1}))}{2\sigma^2}\right) \quad (25)$$

Let $d_i^c = (f_{t\text{-}1}^i - p^c)E_{t\text{-}1}(f_{t\text{-}1}^i - p^c)^T$ be the distance according the EFM between current task feature $f_i$ extracted from the model trained at task $t-1$ and the prototype $p^c$. However, since $d_i^c$ is not bounded (similar to the KL-Divergence), the exponential function becomes unstable across incremental learning steps. To address this problem, we normalize all the distances $d_i^c$ be fall in the interval $[0,1]$ before applying the Gaussian kernel. In all our experiments, we use a fixed $\sigma = 0.2$ for the Gaussian kernel.

## D    PSEUDO-CODE FOR ELASTIC FEATURE CONSOLIDATION

The training procedure for the proposed method is described in Algorithm 1. The algorithm assumes that the first training has already been completed, as it does not require particular treatment.

---

**Algorithm 1:** Elastic Feature Consolidation

**Data:** $\mathcal{M}_1, E_1, \mathcal{P}_1$
**for** $t = 2, \dots T$ **do**
$\quad$ /* Iterate over epochs $\qquad\qquad\qquad\qquad\qquad\qquad\qquad\qquad\qquad\qquad\qquad\qquad$ */
$\quad$ **for** *each optimization step* **do**
$\quad\quad$ sample $x_t \sim \mathcal{X}_t$, $\hat{x}_t \sim \mathcal{X}_t$
$\quad\quad$ sample gaussian proto $\tilde{p} \sim \mathcal{P}_{1:t\text{-}1}$
$\quad\quad$ compute $\mathcal{L}_t^{\text{PR-ACE}}$   (Eq. 12)
$\quad\quad$ compute $\mathcal{L}_t^{\text{EFM}}$   (Eq. 9)
$\quad\quad$ $\mathcal{L}_t^{\text{EFC}} \leftarrow \mathcal{L}_t^{\text{PR-ACE}} + \mathcal{L}_t^{\text{EFM}}$
$\quad\quad$ Update $\theta_t \leftarrow \theta_t - \alpha \frac{\partial \mathcal{L}_t^{\text{EFC}}}{\partial \theta_t}$
$\quad$ **end**
$\quad$ $\mathcal{P}_{1:t\text{-}1} \leftarrow \mathcal{P}_{1:t\text{-}1} + \Delta(E_{t\text{-}1})$   (Eqs. 13, 14)
$\quad$ $\mathcal{P}_{1:t} \leftarrow \mathcal{P}_{1:t\text{-}1} \cup \mathcal{P}_t$
$\quad$ $E_t \leftarrow \text{EFM}(\mathcal{X}_t, \mathcal{M}_t)$   (Eq. 7)
**end**

---

## E    DATASET, IMPLEMENTATION AND HYPERPARAMETER SETTINGS

In this section we expand the content of Section 5.1 and Section 5.2 providing additional information about the dataset and the implementation of the approaches we tested.

### E.1    DATASETS

We perform our experimental evaluation on three standard datasets. CIFAR-100 (Krizhevsky et al., 2009) consists of 60,000 images divided into 100 classes, with 600 images per class (500 for training

and 100 for testing). ImageNet-1K is the original ImageNet dataset (Deng et al., 2009), consisting of about 1.3 million images divided into 1,000 classes. Tiny-ImageNet (Wu et al., 2017) consists of 100,000 images divided into 200 classes, which are taken from ImageNet and resized to $64 \times 64$ pixels. ImageNet-Subset is a subset of the original ImageNet dataset that consists of 100 classes. The images are resized to $224 \times 224$ pixels.

## E.2 Implementation and Hyperparameters

For all the methods, we use the standard ResNet-18 (He et al., 2015) backbone trained from scratch.

**First Task Optimization Details**. We use the same optimization settings for both the Warm Start and Cold Start scenarios. We train the models on CIFAR-100 and on Tiny-ImageNet for 100 epochs with Adam Kingma & Ba (2014) using an initial learning rate of $1e^{-3}$ and fixed weight decay of $2e^{-4}$. The learning rate is reduced by a factor of 0.1 after 45 and 90 epochs (as done in Zhu et al. (2021b); Toldo & Ozay (2022)). For ImageNet-Subset, we followed the implementation of PASS Zhu et al. (2021b), fixing the number of epochs at 160, and used Stochastic Gradient Descent with an initial learning rate of 0.1, momentum of 0.9, and weight decay of $5e^{-4}$. The learning rate was reduced by a factor of 0.1 after 80, 120, and 150 epochs. We applied the same label and data augmentation (random crops and flips) for all the evaluated datasets. For the first task of each dataset, we use self-rotation as performed by Zhu et al. (2021b); Toldo & Ozay (2022).

**Incremental Steps**. Below we provide the hyperparameters and the optimization settings we used for the incremental steps of each state-of-the-art method we tested.

- **EWC (Kirkpatrick et al., 2017)**: We used the implementation of Masana et al. (2022). Specifically, we configured the coefficient associated to the regularizer as $\lambda_{\text{E-FIM}} = 5000$ and the fusion of the old and new importance weights is done with $\alpha = 0.5$. For the incremental steps we fix the total number of epochs to 100 and we use Adam optimizer with an initial learning rate of $1e^{-3}$ and fixed weight decay of $2e^{-4}$. The learning rate is reduced by a factor of 0.1 after 45 and 90 epochs.

- **LwF (Li & Hoiem, 2017)**: We used the implementation of Masana et al. (2022). In particular, we set the temperature parameter $T = 2$ as proposed in the original work and the parameter associated to the regularizer $\lambda_{\text{LwF}}$ to 10. For the incremental steps we fix the total number of epochs to 100 and we use Adam optimizer with an initial learning rate of $1e^{-3}$ and fixed weight decay of $2e^{-4}$. The learning rate is reduced by a factor of 0.1 after 45 and 90 epochs.

- **PASS (Zhu et al., 2021b)**: We follow the implementation provided by the authors. It is an approach relying upon feature distillation and prototypes generation. Following the original paper we set $\lambda_{\text{FD}} = 10$ and $\lambda_{\text{pr}} = 10$. In the original code, we find a temperature parameter, denoted as $T$, applied to the classification loss, which we set to $T = 1$. As provided in the original paper, for the incremental steps we fix the total number of epochs to 100 and we use Adam optimizer with an initial learning rate of $1e^{-3}$ and fixed weight decay of $2e^{-4}$. The learning rate is reduced by a factor of 0.1 after 45 and 90 epochs

- **SSRE (Zhu et al., 2022)**: We follow the implementation provided by the authors. It is an approach relying upon feature distillation and prototypes generation. Following the original paper, we set $\lambda_{\text{FD}} = 10$ and $\lambda_{\text{pr}} = 10$. In the original code, we find a temperature parameter, denoted as $T$, applied to the classification loss, which we set to $T = 1$. Following the original code, for the incremental steps we fixed the total number of epochs to 60 and used Adam Optimizer with an initial learning rate of $2e^{-4}$ and fixed weight decay of $5e^{-4}$. The learning rate is reduced by a factor of 0.1 after 45 epochs.

- **FeTrIL (Petit et al., 2023)**: We follow the official code provided by the authors. During the incremental steps, it uses a Linear SVM classifier working on the pseudo-features extracted from the frozen backbone after the first task. We set the SVM regularization $C = 1$ and the tolerance to 0.0001 as provided by the authors.

**Symmetric loss hyper-parameter**. We fix the value of $\lambda_{\text{pr}} = 10$ in Eq. 11 as reported in the literature (Zhu et al., 2022; 2021b; Toldo & Ozay, 2022).

**Feature Distillation loss hyper-parameter**. Feature distillation can be easily obtained setting $\eta = 10$ and $\lambda_{EFM} = 0$ in our training loss (see Eq. 9). In our preliminary tests, we verified that feature

|  | CIL Method | Warm Start | |
|---|---|---|---|
|  |  | $A_{\text{step}}^K$ 5 Step | $A_{\text{inc}}^K$ 5 Step |
| CIFAR-100 | EWC (Kirkpatrick et al., 2017) | $30.47 \pm 3.38$ | $50.08 \pm 1.13$ |
|  | LwF-MC (Li & Hoiem, 2017) | $43.57 \pm 1.93$ | $59.68 \pm 0.97$ |
|  | PASS (Zhu et al., 2021b) | $56.73 \pm 0.44$ | $65.94 \pm 0.68$ |
|  | Fusion (Toldo & Ozay, 2022) | 58.72 | 66.80 |
|  | FeTrIL (Petit et al., 2023) | $57.52 \pm 0.61$ | $66.15 \pm 0.67$ |
|  | SSRE (Zhu et al., 2022) | $57.79 \pm 0.52$ | $66.53 \pm 0.60$ |
|  | **EFC** | $\mathbf{62.57} \pm 0.33$ | $\mathbf{69.51} \pm 0.64$ |
| Tiny-ImageNet | EWC (Kirkpatrick et al., 2017) | $10.64 \pm 0.59$ | $27.51 \pm 0.61$ |
|  | LwF-MC (Li & Hoiem, 2017) | $39.93 \pm 0.48$ | $52.41 \pm 0.61$ |
|  | PASS (Zhu et al., 2021b) | $44.80 \pm 0.35$ | $53.98 \pm 0.48$ |
|  | Fusion (Toldo & Ozay, 2022) | 48.57 | - |
|  | FeTrIL (Petit et al., 2023) | $46.35 \pm 0.28$ | $54.96 \pm 0.39$ |
|  | SSRE (Zhu et al., 2022) | $44.87 \pm 0.45$ | $54.23 \pm 0.31$ |
|  | **EFC** | $\mathbf{51.19} \pm 0.49$ | $\mathbf{58.12} \pm 0.53$ |

Table A1: Performance in the 5-step Warm Start scenario on CIFAR-100 and Tiny-Imagenet.

distillation is robust with respect to this hyper-parameter, especially in Warm Start Scenario, due to little plasticity introduced in the incremental steps.

## F  CIFAR-100 AND TINY-IMAGENET PERFORMANCE ON 5-STEP WARM START

Many methods are evaluated using short task sequences. To facilitare broad comparison with the state-of-the-art, in Table A1 we report performance of all implemented methods in the 5-Step Warm Start scenario on CIFAR-100 and Tiny-ImageNet.

## G  EXPERIMENTAL RESULTS ON IMAGENET-1K

To comprehensively evaluate the performance of EFC, we conducted experiments across all analyzed scenarios on ImageNet-1K (see Table A2). In the Cold Start scenario, we evenly split the classes, with 100 for the 10-step scenario and 50 for the 20-step scenario. In the Warm Start scenario, we consider a large initial task with 500 classes for the 10-step scenario and 400 classes for the 20-step scenario. We compare our results with FeTrIL, which exhibited performance close to ours on all other datasets. For these experiments, we use the same hyperparameters and training protocol as in the ImageNet-Subset scenario in Appendix E.2, but we fix the batch size to 256 and deactivate self-rotation for the first task. Furthermore, we fix the class ordering to align with that reported by Yan et al. (2021).

Even on the much larger ImageNet-1K dataset, our results demonstrate that in both Warm and Cold Start scenarios EFC remains highly effective at maintaining plasticity throughout training. Specifically, in the Warm Start scenario EFC exhibits a substantial improvement of approximately 4% over FeTrIL in both per-step and average incremental accuracy. It is worth noting that the results reported in Petit et al. (2023) are slightly better in the Warm Start scenario in terms of average incremental accuracy with respect to those reported in Table A2. This gap can probably be attributed to the performance obtained on the initial task, which has a strong impact in terms of average incremental accuracy. In Cold Start scenarios, we observe once again that freezing the feature extractor does not provide enough plasticity for FeTrIL to learn the new tasks effectively, resulting in a significant decrease (more than 8%) in accuracy compared to EFC.

| Method | Warm Start | | | | Cold Start | | | |
|---|---|---|---|---|---|---|---|---|
|  | $A_{\text{step}}^K$ | | $A_{\text{inc}}^K$ | | $A_{\text{step}}^K$ | | $A_{\text{inc}}^K$ | |
|  | 10 Step | 20 Step | 10 Step | 20 Step | 10 Step | 20 Step | 10 Step | 20 Step |
| FeTrIL | $55.26 \pm 0.16$ | $51.10 \pm 0.22$ | $63.42 \pm 0.08$ | $60.93 \pm 0.12$ | $34.28 \pm 0.20$ | $26.64 \pm 0.25$ | $48.93 \pm 0.15$ | $40.02 \pm 0.15$ |
| **EFC** | $\mathbf{59.49} \pm 0.07$ | $\mathbf{55.94} \pm 0.09$ | $\mathbf{66.64} \pm 0.10$ | $\mathbf{64.89} \pm 0.10$ | $\mathbf{42.62} \pm 0.08$ | $\mathbf{36.32} \pm 0.35$ | $\mathbf{56.52} \pm 0.10$ | $\mathbf{49.80} \pm 0.20$ |

Table A2: Performance in the 10-step and 20-step in Warm and Cold Start scenarios on ImageNet-1K.

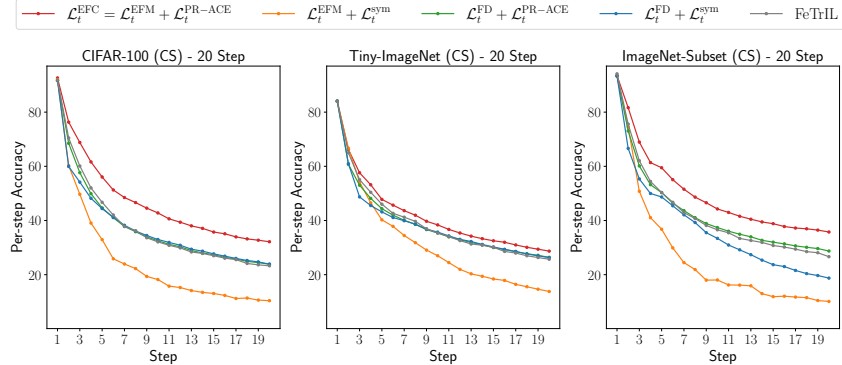

Figure A4: Ablation on regularization and prototype losses.

| Dataset | Step | Update Proto (WS) | | Update Proto (CS) | |
|---|---|---|---|---|---|
| | | ✗ | ✓ | ✗ | ✓ |
| CIFAR-100 | 10 | $58.24 \pm 0.62$ | $\mathbf{60.87} \pm 0.39$ | $39.28 \pm 0.97$ | $\mathbf{43.62} \pm 0.70$ |
| | 20 | $52.60 \pm 0.73$ | $\mathbf{55.78} \pm 0.42$ | $27.28 \pm 1.48$ | $\mathbf{32.15} \pm 1.33$ |
| Tiny-ImageNet | 10 | $48.16 \pm 0.53$ | $\mathbf{50.40} \pm 0.25$ | $31.23 \pm 0.88$ | $\mathbf{34.10} \pm 0.77$ |
| | 20 | $45.70 \pm 0.74$ | $\mathbf{48.68} \pm 0.65$ | $25.12 \pm 0.23$ | $\mathbf{28.69} \pm 0.40$ |
| ImageNet-Subset | 10 | $66.92 \pm 0.43$ | $\mathbf{68.85} \pm 0.58$ | $39.99 \pm 2.20$ | $\mathbf{47.38} \pm 1.43$ |
| | 20 | $59.90 \pm 1.09$ | $\mathbf{62.17} \pm 0.69$ | $29.62 \pm 2.60$ | $\mathbf{35.75} \pm 1.74$ |

Table A3: Ablation on prototype update.

## H ADDITIONAL ABLATION STUDIES

In this section we extend the ablation study in 5.3 by providing the same analysis on the Tiny-ImageNet and ImageNet-Subset datasets.

Considering first the the ablation on PR-ACE and Feature Distillation, Figure A4 confirms that our asymmetric loss is a pivotal component in controlling the plasticity introduced by the EFM loss, surpassing feature distillation.

For the ablation on prototype updates, Table A3 clearly shows that updating prototypes enhances performance across all datasets and scenarios. As discussed in the main paper, the performance improvement is more significant in the Cold Start scenario, where feature drift is more pronounced compared to the Warm Start scenario.

## I ANALYSIS OF COMPUTATION AND STORAGE COST

In this section we consider the computational and storage efficiency of EFC, which computes the EFM after each task, regularizes using the EFM pseudo-metric, and stores class covariance matrices for Gaussian prototype sampling. In terms of storage efficiency, a spectral analysis of the covariance matrices reveals that for all classes in all datasets and all task sequences the spectra are highly concentrated in the first 30-50 eigenvalues and are thus well-approximated by low-rank reconstructions. To verify this we ran experiments with low-rank reconstructions and give results for CIFAR-100 and Tiny-ImageNet in Table A4. On average, using only 30 eigenvectors is sufficient to preserve 99% of the total variance and maintain the same EFCIL accuracy.

In Table A5 we consider two more aggressive strategies for mitigating storage costs. The **Each Task** approaches uses a single covariance matrix *per task* for prototype generation and requires storage scaling linearly in number of tasks, while the **Last Task** stores only the covariance matrix from the last task learned and thus requires *constant* storage. Although the EFCIL accuracy suffers from using these proxies, the EFC results are still comparable with the state-of-the-art. Specifically, in the Warm Start scenario storing one covariance per class has a significant impact since the covariance matrices slightly change across the incremental learning steps. With respect to the Cold start scenario in this specific setting, feature drift is mitigated by the fact that we learn half of all available classes during the first session and thus the initial representation are already effective for new tasks. On the

| | **CIFAR-100** | | | **Tiny-ImageNet** | | |
|---|---|---|---|---|---|---|
| | Preserved Variance | Avg # Components | $A^K_{step}$ | Preserved Variance | Avg # Components | $A^K_{step}$ |
| 5 Step | 0.90 | $11 \pm 2$ | 57.35 | 0.90 | $15 \pm 3$ | 49.02 |
| | 0.95 | $18 \pm 4$ | 59.68 | 0.95 | $23 \pm 4$ | 49.72 |
| | 0.99 | $39 \pm 6$ | 61.65 | 0.99 | $48 \pm 6$ | 50.85 |
| | 1.00 | 512 | 62.57 | 1.00 | 512 | 51.19 |
| 10 Step | 0.90 | $11 \pm 2$ | 55.79 | 0.90 | $15 \pm 3$ | 47.79 |
| | 0.95 | $18 \pm 3$ | 58.28 | 0.95 | $24 \pm 4$ | 48.77 |
| | 0.99 | $38 \pm 6$ | 60.66 | 0.99 | $48 \pm 6$ | 50.54 |
| | 1.00 | 512 | 60.87 | 1.00 | 512 | 50.40 |
| 20 Step | 0.90 | $11 \pm 2$ | 50.67 | 0.90 | $15 \pm 3$ | 46.64 |
| | 0.95 | $18 \pm 3$ | 52.95 | 0.95 | $23 \pm 4$ | 47.49 |
| | 0.99 | $38 \pm 6$ | 54.91 | 0.99 | $48 \pm 6$ | 48.39 |
| | 1.00 | 512 | 55.78 | 1.00 | 512 | 48.68 |

Table A4: Warm Start EFC performance with low-rank covariance approximations for Gaussian prototype sampling. On both datasets in all task sequences using fewer than 10% of the principal directions on average preserves 99% of the total variance and maintains the same accuracy as using full covariance matrices (full covariance with preserved variance = 1.00 given for reference).

| | | **Warm Start** | | | | **Cold Start** | | | |
|---|---|---|---|---|---|---|---|---|---|
| | | $A^K_{step}$ | | $A^K_{inc}$ | | $A^K_{step}$ | | $A^K_{inc}$ | |
| | Variant | 10 Step | 20 Step | 10 Step | 20 Step | 10 Step | 20 Step | 10 Step | 20 Step |
| CIFAR-100 | Each Task | $59.37 \pm 0.18$ | $54.97 \pm 0.86$ | $67.34 \pm 0.38$ | $65.12 \pm 0.92$ | $\mathbf{45.26} \pm 0.62$ | $\mathbf{33.73} \pm 2.38$ | $\mathbf{60.11} \pm 0.85$ | $\mathbf{48.96} \pm 1.90$ |
| | Last Task | $58.22 \pm 0.80$ | $52.79 \pm 0.10$ | $66.79 \pm 0.49$ | $63.89 \pm 0.86$ | $43.72 \pm 0.38$ | $31.43 \pm 2.48$ | $59.46 \pm 0.99$ | $47.27 \pm 1.53$ |
| | **EFC** | $\mathbf{60.87} \pm 0.39$ | $\mathbf{55.78} \pm 0.42$ | $\mathbf{68.23} \pm 0.68$ | $\mathbf{65.90} \pm 0.97$ | $43.62 \pm 0.70$ | $32.15 \pm 1.33$ | $58.58 \pm 0.91$ | $47.36 \pm 1.37$ |
| Tiny-ImgNet | Each Task | $49.00 \pm 0.48$ | $47.26 \pm 0.18$ | $56.31 \pm 0.51$ | $55.06 \pm 0.39$ | $\mathbf{34.81} \pm 0.56$ | $\mathbf{29.39} \pm 0.52$ | $47.84 \pm 0.39$ | $\mathbf{42.53} \pm 0.17$ |
| | Last Task | $48.00 \pm 0.43$ | $45.64 \pm 0.51$ | $55.85 \pm 0.54$ | $54.12 \pm 0.37$ | $34.36 \pm 0.14$ | $28.57 \pm 0.32$ | $47.50 \pm 0.24$ | $42.11 \pm 0.11$ |
| | **EFC** | $\mathbf{50.40} \pm 0.25$ | $\mathbf{48.68} \pm 0.65$ | $\mathbf{57.52} \pm 0.43$ | $\mathbf{56.52} \pm 0.53$ | $34.10 \pm 0.77$ | $28.69 \pm 0.40$ | $\mathbf{47.95} \pm 0.61$ | $42.07 \pm 0.96$ |
| ImgNet-Sub. | Each Task | $66.53 \pm 0.48$ | $60.55 \pm 0.73$ | $74.10 \pm 0.62$ | $70.36 \pm 1.24$ | $\mathbf{48.49} \pm 2.18$ | $\mathbf{36.31} \pm 3.35$ | $\mathbf{61.19} \pm 2.46$ | $\mathbf{50.52} \pm 2.22$ |
| | Last Task | $65.46 \pm 0.69$ | $59.20 \pm 0.69$ | $73.42 \pm 0.72$ | $69.59 \pm 1.40$ | $47.21 \pm 1.57$ | $35.43 \pm 3.00$ | $60.50 \pm 2.16$ | $50.22 \pm 1.94$ |
| | **EFC** | $\mathbf{68.85} \pm 0.58$ | $\mathbf{62.17} \pm 0.69$ | $\mathbf{75.40} \pm 0.92$ | $\mathbf{71.63} \pm 1.13$ | $47.38 \pm 1.43$ | $35.75 \pm 1.74$ | $59.94 \pm 1.38$ | $49.92 \pm 2.05$ |

Table A5: Mitigating storage costs with proxy covariance matrices. We can save only one covariance matrix per task (**Each Tas**k) to use for prototype generation, which results in storage that grows linearly in number of tasks. Alternatively, we can store only the covariance matrix from the **Last Task**, which results in a constant storage cost. Both proxies sacrifice some accuracy in the Warm Start scenario, but results are still comparable with the state-of-the-art (compare with results in Table 1).

contrary, in Cold Start scenario the representations are more prone to changes, and having per-class fixed covariances is less crucial. Consequently, the differences in Cold Start Scenario between the variants of our method are practically negligible. Theoretically, both scenarios, but especially Cold Start, could be improved if we design a method to update these matrices during incremental learning. However, as we highlight in Section 6) of the paper, this remains an open point.

In terms of computational efficiency we derived a method (Eq. 7 in the main paper and Appendix B.1) for computing the EFM without backward operations. Finally, we conducted a wall-clock time analysis to assess the computational costs of EFC. The results of this analysis are presented in Figure A5. We see that EFC does not add significant overhead, but rather it is the addition of prototypes or exemplars that yields the most significant increase in computational time. LwF is the fastest methods since it does not rely upon prototype rehearsal. Among the prototype-based methods, EFC is comparable to SSRE in terms of computational time and it is much faster than PASS which is more time-consuming since applies self-rotation across the incremental steps. In the plot we do not report FeTrIL, relying upon SVM for the training. Currently, FeTrIL is the fastest state-of-the-art EFCIL approach since it freezes the backbone and updates only the classifier.

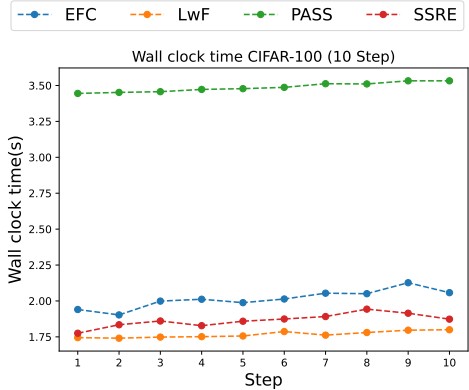

Figure A5: Wall clock timing comparison. We compute the per-epoch time averaged over 100 epochs.

## J    WARM START PER-STEP ACCURACY PLOTS

In Figure A6 we compare the per-step incremental accuracy of EFC and recent approaches on Warm Start scenarios in all three datasets for various task sequences. All methods begin from the same starting point after training on the large first task, but EFC maintains more plasticity thanks to the EFM regularizer and is thus better at learning new tasks.

## K    COLD START PER-STEP ACCURACY PLOTS

In Figure A7 we compare the per-step incremental accuracy of EFC and recent approaches on Cold Start scenarios in all three datasets for various task sequences. All methods begin from the same starting point after training on the small first task, but EFC both maintains plasticity thanks to the EFM regularizer and is able to better update previous task classifiers to learn new tasks and mitigate forgetting.

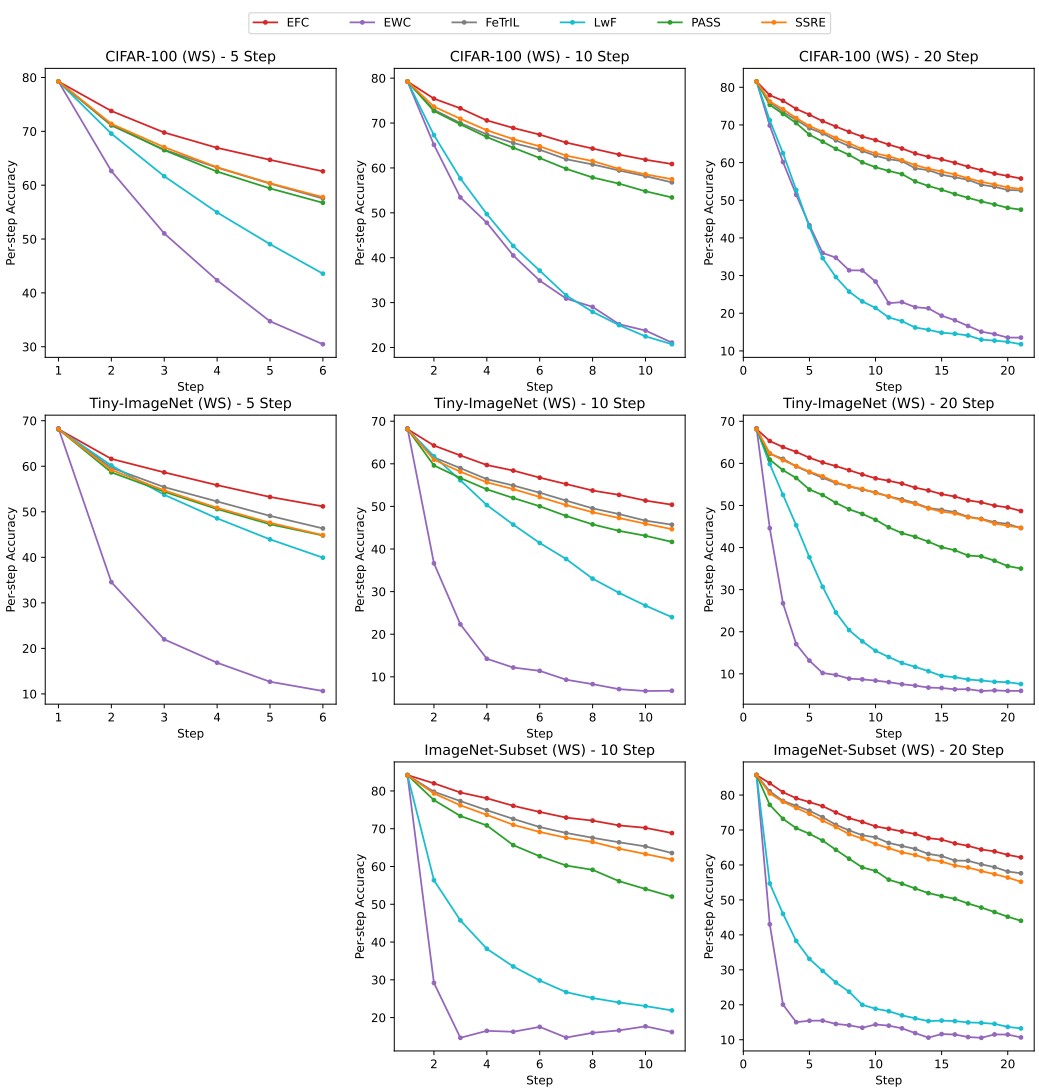

Figure A6: Warm Start (WS) per-step accuracy during the incremental learning. The plots compare recent EFCIL methods against EFC on three different dataset for different incremental step sequences.

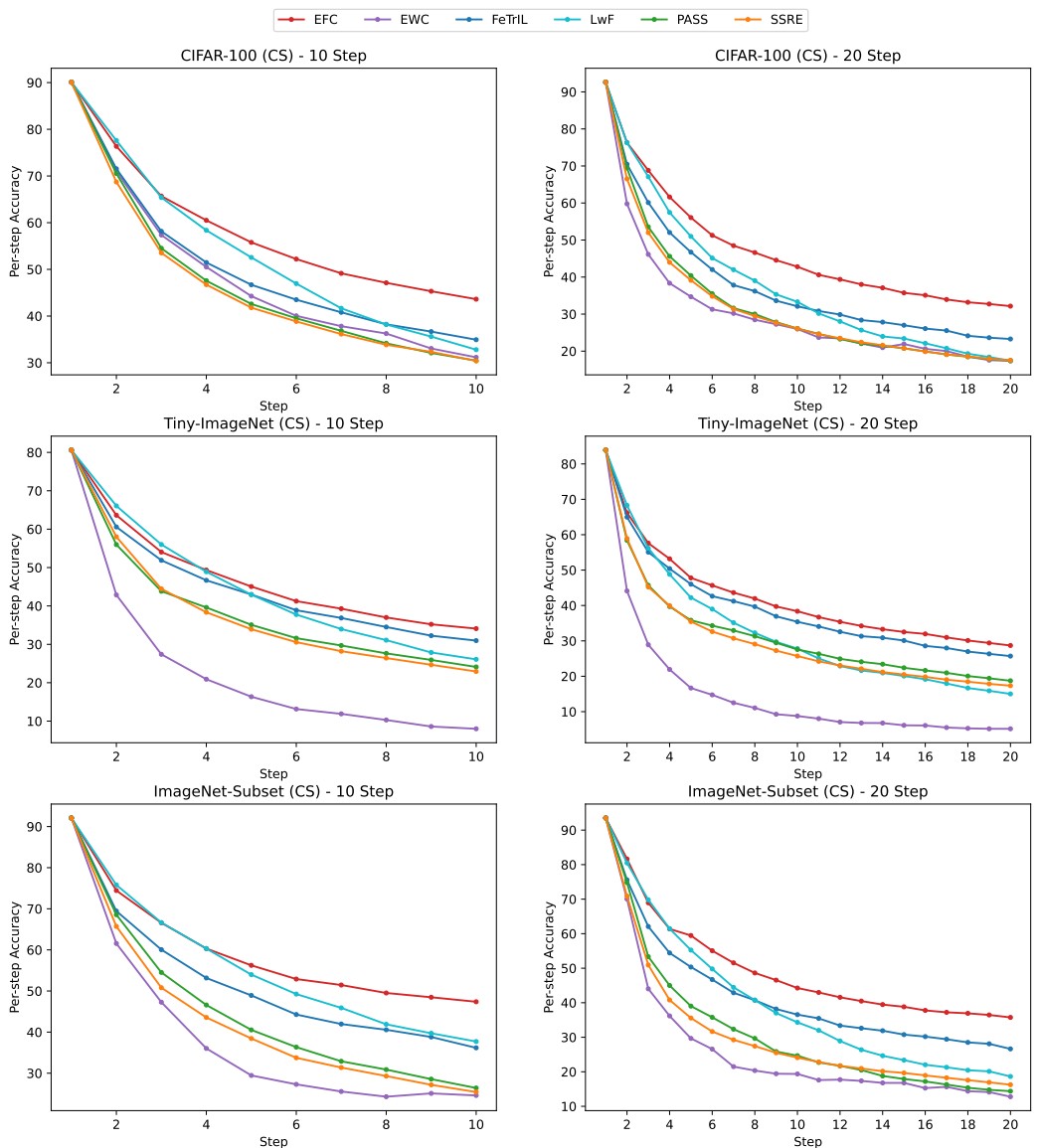

Figure A7: Cold Start (CS) per-step accuracy plots during incremental learning. The plots compare recent EFCIL methods with EFC on three different datasets for different incremental step sequences.

