# OpenReview forum: "Elastic Feature Consolidation For Cold Start Exemplar-Free Incremental Learning"
_ICLR.cc/2024/Conference — ICLR 2024 poster_

### Official Review · Reviewer_RQMN · 2023-10-30

**Soundness:** 3 good
**Presentation:** 3 good
**Contribution:** 3 good
**Rating:** 6
**Confidence:** 5

**Summary:**

This paper studies the examplar-free class incremental learning (EFCIL) problem. The author is motivated by the cold-start scenario, i.e., the first task is insufficiently large to learn a high-quality backbone and thus demands the method to be more plastic to adapt their backbone to learn new tasks. Moreover, the author thinks existing methods in EFCIL pose too strong a regularization to restrict the model to be plastic, like the EWC in parameter space and the L2 regularization in feature space. To give the model more plasticity in learning new tasks, the author proposes a new method called the empirical feature matrix (EFM), which is a variant of the empirical fisher matrix in that the author calculates the square of the derivative over the feature extractor instead of the whole network and uses this as a regularization term to penalize the change in feature space. Moreover, the author combines both the prototype rehearsal and prototype drift compensation from existing works via the proposed EFM to create an asymmetric prototype rehearsal loss to provide elastic feature consolidation. Extensive experiments are conducted on the standard continual learning benchmarks and the author also compares with state-of-the-art EFCIL methods and provides further analysis.

**Strengths:**

1. Overall, the paper is well-written. The idea of the empirical feature matrix is an intuitive extension of the empirical fisher matrix, which seems interesting. The combination of the existing methods of prototype rehearsal and prototype drift compensation makes the proposed methods more sound given that the effectiveness of these components has been empirically verified previously.
2. The proposed method can achieve new SOTA performance on either warm or cold start scenarios, which demonstrates the effectiveness and generality of the proposed method.

**Weaknesses:**

1. The reviewer is unconvinced with the so-called definition of the "cold start" scenario in the present paper. In the definition provided by the author in the Introduction:

"In this paper we consider EFCIL in the more challenging Cold Start scenario in which the first task is insufficiently large to learn a high-quality backbone and methods must be plastic and adapt their backbone to new tasks."

First, how to quantify the term "insufficiently large"? The reviewer sees that the author uniformly distributes all classes among all tasks in the Experiments, which means like 10 or 5 classes learned in step 1 for CIFAR-100. However, such experimental protocols are widely used in existing CL literature (e.g., see [1] and its follow-up works), which should be a *standard* experimental protocol for benchmarking CL. Moreover, how can the author confirm that 10 classes can not be a "sufficiently large" task? What criteria that the author explicitly measure whether a task is "sufficient" or "insufficiently" large?

Second, how can the author confirm that the backbone is not of "high quality"? What quality does the author refer to? The quality of learning the current tasks or the quality of learning the new tasks? Why does the model need to have more plasticity for learning the new task if the model is not of "high quality"? One can also say that the model has **exactly high plasticity** for learning new tasks given that the model has **only** learned for 5-10 classes given the overparameterized property of the modern CNN/DNN.

Overall, the definition of the so-called "cold-start" scenario and the corresponding motivations that lead the author to propose the current method are not solid and sound. This paper may still stand out even if we do not mention the so-called "cold-start" scenario at first and instead only talk about the strong regularization effect made by existing methods and thus demand more flexible regularization methods. Then what is the necessity of this "cold-start" scenario?

Last but not least, from the convention in machine learning, cold-start scenarios often refer to scenarios where we do not have any data to start with, e.g., for the recommendation system. With 5-10 classes (which means thousands of images) for training the model can hardly be thought of as a "cold" start.

2. The details of Figure 2 are missing. What dataset and experimental protocols have the author used? Will the conclusion be influenced by the number of classes? Will the number of training data in the initial task have an impact on the conclusion? The author should provide the mean and std and repeat experiments to support this empirical observation.

Minor:
1. Typo in the first sentence of Section 5. "with with" -> "with"


[1] DER: Dynamically Expandable Representation for Class Incremental Learning (CVPR 2021)

**Questions:**

Please refer to the Weaknesses section for more details.

---

> ### Author Response · Authors · 2023-11-18
> **Response 1/2**
>
> We thank the reviewer for their insightful comments and constructive criticism of our work. Below reply to specific issues raised, with particular attention to the central one regarding our definition of and motivations for the Cold Start scenario in class-incremental learning.
>
> ---
> > The reviewer is unconvinced with the so-called definition of the "cold start" scenario in the present paper. In the definition provided by the author in the Introduction:
> "In this paper we consider EFCIL in the more challenging Cold Start scenario in which the first task is insufficiently large to learn a high-quality backbone and methods must be plastic and adapt their backbone to new tasks"
>
> > First, how to quantify the term "insufficiently large"? The reviewer sees that the author uniformly distributes all classes among all tasks in the Experiments, which means like 10 or 5 classes learned in step 1 for CIFAR-100. However, such experimental protocols are widely used in existing CL literature (e.g., see [1] and its follow-up works), which should be a standard experimental protocol for benchmarking CL. Moreover, how can the author confirm that 10 classes can not be a "sufficiently large" task? What criteria that the author explicitly measure whether a task is "sufficient" or "insufficiently" large?
>
> The reviewer is correct in pointing out that the Cold Start scenario we consider is a standard protocol for *exemplar-based* methods. However, for *exemplar-free* methods, which we study exclusively in this paper, this Cold Start scenario is not used and instead a Warm Start consisting of a *large first task* is used [D, E, F, G]. For exemplar-free methods backbone drift leads to catastrophic forgetting, and therefore methods allow very little plasticity in the backbone during incremental learning. Our paper is *one of the first to consider Cold Start scenarios for exemplar-free class-incremental learning*.
>
> [D] Prototype augmentation and self-supervision for incremental learning (CVPR 2021), Zhu et al.
>
> [E] Fetril: Feature translation for exemplar-free class-incremental learning (WACV 2023), Petit et al.
>
> [F] Bring evanescent representations to life in lifelong class incremental learning (CVPR 2022), Toldo et al.
>
> [G] Self-sustaining representation expansion for non-exemplar class-incremental learning (CVPR 2022), Zhu et al.
>
> ---
> > Second, how can the author confirm that the backbone is not of "high quality"? What quality does the author refer to? The quality of learning the current tasks or the quality of learning the new tasks? Why does the model need to have more plasticity for learning the new task if the model is not of "high quality"? One can also say that the model has exactly high plasticity for learning new tasks given that the model has only learned for 5-10 classes given the overparameterized property of the modern CNN/DNN.
>
> Given the feedback of the reviewer, we think it is better to adopt a more practical definition of Cold Start. A Cold Start scenario for a dataset is when *training the backbone on the first task it is of considerably lower quality than when training on half of the dataset*. We performed an experiment on CIFAR-100 to quantify this by incrementally training a ResNet-18 backbone and testing it via linear probing (i.e. training a linear classifier with a cross-entropy loss) on an independent set of 50 classes. The results of this are given in the following table.
>
> **Linear Probing (CIFAR-100)**:
> | # of Base Classes | Linear Probe Accuracy (%) |
> |----| ---- |
> |10  | 49.8 |
> |20  | 59.7 |
> |30  | 62.0 |
> |40  | 64.8 |
> |50  | 67.9 |
>
> The backbone trained on 50 classes outperforms the one trained on 10 classes by more than 18%. In any incremental learning scenario, the backbone starting from only 10 classes will require more plasticity to achieve high incremental learning accuracy. Freezing the backbone (e.g. FeTrIL) or feature distillation do not leave enough plasticity in learning to acquire and perform well on new tasks.
>
> We feel that it is important to distinguish model *capacity* from learning *plasticity*. When a model is trained on a small first task it is true that the model retains a large *capacity* to learn new tasks, however without adequate *plasticity* when learning new tasks -- which is a property of the *learning method* and not only of the *model* itself -- this capacity cannot be effectively exploited because the backbone is not allowed the plasticity needed to exploit its available capacity.
>
> Our experimental results, ablations, and analyses we provide show that Elastic Feature Consolidation is very effective at manging this additional plasticity needed to acquire new tasks in both Cold Start and Warm Start scenarios. We feel that this is an important and novel contribution since, until now, the emphasis has been almost entirely focused on Warm Start scenarios for exemplar-free continual learning -- scenarios in which backbone plasticity is significantly less important.
>
> **Response 1/2**

---

> ### Author Response · Authors · 2023-11-18
> **Response 2/2**
>
> > Overall, the definition of the so-called "cold-start" scenario and the corresponding motivations that lead the author to propose the current method are not solid and sound. This paper may still stand out even if we do not mention the so-called "cold-start" scenario at first and instead only talk about the strong regularization effect made by existing methods and thus demand more flexible regularization methods. Then what is the necessity of this "cold-start" scenario?
>
> It is important to distinguish Cold from Warm Start scenarios precisely because that is a characteristic that distinguishes scenarios requiring plasticity in learning (and thus effective and targeted regularization) from those that do not. Our aim is to give names to these two scenarios and to emphasize the importance of treating them differently.
>
> ---
>
> > Last but not least, from the convention in machine learning, cold-start scenarios often refer to scenarios where we do not have any data to start with, e.g., for the recommendation system. With 5-10 classes (which means thousands of images) for training the model can hardly be thought of as a "cold" start.
>
> We recognize the point the reviewer makes, but at the same time believe that our experimental comparison with the state-of-the-art in our Cold Start scenarios illustrates that the two types of scenarios indeed need to be handled quite differently in class-incremental learning. Moreover, from a terminological point of view it is very useful to distinguish scenarios beginning with a first task significantly larger than the following tasks (Warm Start) from those that do not (Cold Start).
>
> These points raised by the reviewer have stimulated much discussion and we recognize that the issue is more nuanced than our treatment in the original submission makes it seem. We will integrate elements of this discussion in the final version of the paper in order to stimulate further discussion on the relationship between model capacity and plasticity in incremental learning, and Warm versus Cold starts for class-incremental learning.
>
> ---
>
> > The details of Figure 2 are missing. What dataset and experimental protocols have the author used? Will the conclusion be influenced by the number of classes? Will the number of training data in the initial task have an impact on the conclusion? The author should provide the mean and std and repeat experiments to support this empirical observation.
>
>
> We agree that some details were missing and thank the reviewer for pointing this out. This analysis is, however, independent of the setting employed (in the paper we use CIFAR-100 - Warm Start - 5 step). To make this more clear we ran our analysis again considering the following settings: CIFAR-100 (WS) - 10 Step, CIFAR-100 (CS) - 10 Step (see the new Appendix J in the updated manuscript). Our findings are confirmed in both scenarios. Indeed, the perturbation in significant directions changes the posterior probability of the model in a significant way (in this case the first task), and the matrix $E_3$ captures all important directions in feature space up through task 3. This demonstrates the independence of our claim from the first task dimension and the number of classes in each incremental step.
>
> To illustrate robustness across different seeds, we also show the average accuracy drop when adding Gaussian perturbations along principal and non-principal directions (separately) with respect to a regular class incremental learning scenario up to the task 3 (also in the new Appendex K, Figure A8). This plot reveals that perturbations along non-principal directions have no impact on accuracy (indicated by three dashed, overlapped lines on the $x$-axis), while perturbations along the significant direction consistently degrade the performance of our models.
>
> **Response 2/2**

---

> > ### Comment · Reviewer_RQMN · 2023-11-20
> >
> > Thanks to the author for providing the detailed response. My concerns about the motivation of the paper have mostly been resolved. Besides the motivation, the present paper has done a good job. Thus, I increase my score accordingly.

---

### Official Review · Reviewer_Hr2j · 2023-10-30

**Soundness:** 3 good
**Presentation:** 3 good
**Contribution:** 3 good
**Rating:** 8
**Confidence:** 4

**Summary:**

This paper investigates the cold start problem in exemplar-free class incremental learning (EFCIL) scenario. In this setup, the first task contains limited classes and the backbone should be plastic enough to learn other classes. The Elastic Feature Consolidation (EFC) is proposed to balance the plasticity and stability in EFCIL. Experimental results demonstrate the superiority of EFC.

**Strengths:**

Although it is commonly considered in exemplar-based class incremental learning, the cold start scenario is ignored in exemplar-free class incremental learning. EFCIL with cold start brings new challenges, and this paper is well-motivated to address them.

Inspired by the well-known Elastic Weight Consolidation (EFC), the proposed Elastic Feature Consolidation (EFC) is novel and effective.

I like the illustration in Figure 2, which demonstrates the motivation of EFC. Besides, the spectral analysis in B.2 is interesting.

**Weaknesses:**

Lack of experiments on Imagenet-1k dataset.

**Questions:**

No more questions

---

> ### Author Response · Authors · 2023-11-18
>
> We thank the reviewer for recognizing that EFC is one of the first methods to deal with the Cold Start scenario, until now largely overlooked in the exemplar-free class incremental literature. We greatly appreciate the acknowledgment of the novelty of our proposed methods and the spectral analysis of the Empirical Feature Matrix. Below we respond to the main question raised by the reviewer.
>
> ---
>
> We performed single runs of 10-Step Cold Start and Warm Start scenarios for EFC on ImageNet-1K and compare with FeTrIL, the closest to ours in terms of performance on the other scenarios. Given the limited time and our limited computational resources, it was not possible to run other baselines on ImageNet-1K at this time.
>
> The following tables give the results for both scenarios and metrics:
>
> **ImageNet-1K 10-step Cold Start**:
> |  | $A^K_{step}$ | $A^K_{inc}$|
> |---|---|---|
> FeTrIL | 33.86 | 48.46
> EFC | **42.35** | **56.14**
>
> **ImageNet-1K 10-step Warm Start**:
> |  | $A^K_{step}$ | $A^K_{inc}$|
> |---|---|---|
> FeTrIL | 54.93 | 63.12
> EFC | **59.34** | **66.36**
>
> For these experiments we used the same hyperparameters and training protocol as on ImageNet-Subset (see Appendix E of the Supplementary Material). Even on the much larger ImageNet-1K dataset, these results show that in both Warm and Cold Start scenarios EFC is still very effective at maintaining plasticity throughout training. EFC is more effective at learning new tasks compared to FeTrIL, which represents the current state-of-the-art in exemplar-free class-incremental learning.
>
> We are continuing to run additional baselines for these scenarios and will include these new results in the final version.

---

> > ### Comment · Reviewer_Hr2j · 2023-11-22
> >
> > I really thank the authors for the extra effort in adding experiments on ImageNet-1K dataset, and I do not have any other concerns about this work. Therefore, I increase my score to a clear acceptance.

---

### Official Review · Reviewer_sY5k · 2023-10-31

**Soundness:** 4 excellent
**Presentation:** 4 excellent
**Contribution:** 3 good
**Rating:** 8
**Confidence:** 5

**Summary:**

This paper is on the topic of continual learning. Specifically, it addresses the popular scenario of Class Incremental Learning (CIL). Within CIL, it assumes two of the most challenges constraints: (1) no exemplar/rehearsal memory (in which samples from past tasks are stored in memory and used to supplement training of new tasks) is permitted (EFCIL) and (2) the size of the first task is not large - the authors call this cold-start.  The paper takes careful note of both main challenges in continual learning: forgetting, and inter-task confusion, and weaknesses in past method, EWC. The new approach proposed aims to address both challenges, and the weakness of EWC: the strategy is to define a loss function that mitigates forgetting (by regularization of features rather than  weights) but still allows for plasticity, to ensure the new task can be learned.  The net result is the loss stated in Equation (15).  From ablation studies on split CIFAR-100, this new loss is sufficient to outperform comparison methods on the cold start scenario by large margins. But the papers adds an addition trick to perform even better: the use of prototype updates.  The new losses are supported by careful theoretical derivations.

**Strengths:**

Originality: the paper makes two novel contributions to EFCIL - the combination new loss (EFM+PR-ACE), and supplementation of this with their prototype rehearsal method.

Quality: the chain of reasoning motivating the approach, and the supporting theory are carefully argued and demonstrate a deep understanding of the challenges in EFCIL, resulting in new SOTA performance in this subset of CIL. The appendices are well written, comprehensive and support the paper well. I looked through the code supplied as SM and found it ran out of the box and reproduced one of the results.

Clarity: the paper is very well expressed and presented. I liked the sequence of material, and found it enjoyable to read.

Significance: this paper has clear importance for EFCIL and is likely to be an often compared-with benchmark for future competing methods. I think there's scope for the approaches here to impact on work that does assume exemplars, and potentially on EFCIl methods that utilise pre-trained models.

**Weaknesses:**

I only have one weakness within the material presented: I would prefer to see ablations for more than just one dataset. In particular, in Fig 4, only CIFAR-100 is seen, in Table 2, only TinyImageNet. Figure 4(b) tells me that the EFM+PR-ACE loss is enough by itself (i.e. without prototype rehearsal) for better results than comparison methods on cold startup. But is this also true for the other datasets?

Since I only have one weakness to list on the contributions of the paper, I'll take the opportunity to make a mild critique of two weaknesses (in my opinion) in the assumptions that are made in the problem setup in this paper, and in much related work in continual learning.

1. Recently some of the community have started to question the frequent assumption that we need CL methods because storage of data is expensive, or unavailable due to privacy. See for example https://arxiv.org/abs/2305.09253
    - I understand there's very little space to discuss the arguments for and against the importance of "exemplar free" but it would be better to at least acknowledge that there's some debate on this.

2. This paper, and many others like it, seek a CL method that trains a network from a random initial state. However, in practical industry settings, it's nearly always better to consider transfer learning, which for an image classifier has for a long time meant starting with a model trained on ImageNet, but nowadays is also meaning self-supervised pre-training. It's puzzling why the CL community ignore this for the most part. This might be changing now, with a bunch of papers in 2023 now doing CIL with pretrained ViT models (and also ResNets - e.g. your cited ref Panos et al). It could be a simple experiment to try this and assess the benefits of using your method in this different assumption. I am raising to highlight a potential opportunity, and to recommend you at least mention your assumption of the need to train a model from scratch rather than use pre-trained weights.

**Questions:**

As per above, do ablations on all data sets show the same trend, i.e. that the EFM+PR-ACE loss is enough to surpass comparison methods by itself, and adding prototypes rehearsal to this creates a further boost?

I found it very difficult to read the tiny font in Figure 2 and Figure 4(a), so please can you find a way to make those more easily readable?

---

> ### Author Response · Authors · 2023-11-18
>
> We thank the reviewer for their kind words and insightful comments, questions and suggestions regarding our work. Many thanks indeed for the appreciation of the organization and clarity of our manuscript, and also of the quality of submitted code. Below we reply to specific comments made.
>
> ---
>
> > I only have one weakness within the material presented: I would prefer to see ablations for more than just one dataset. In particular, in Fig 4, only CIFAR-100 is seen, in Table 2, only TinyImageNet. Figure 4(b) tells me that the EFM+PR-ACE loss is enough by itself (i.e. without prototype rehearsal) for better results than comparison methods on cold startup. But is this also true for the other datasets?
>
> The complete version of Table 2 was already in Table A3 of the Supplementary Material. We have added a new Appendix J that provides the requested ablations on ImageNet-Subset and TinyImageNet. We see that incorporating prototype updates in certain cases in the Cold Start Scenario allows us to surpass the performance of existing methods, something that does not always happen when using fixed prototypes. For example, FeTrIL (CS) Tiny-ImageNet - 20 step reaches 25.70% in accuracy compared to only 25.12% achieved by EFC with fixed prototypes.
>
> ---
>
> > Recently some of the community have started to question the frequent assumption that we need CL methods because storage of data is expensive, or unavailable due to privacy. See for example https://arxiv.org/abs/2305.09253. I understand there's very little space to discuss the arguments for and against the importance of "exemplar free" but it would be better to at least acknowledge that there's some debate on this.
>
> We agree and have also noticed the debate heating up on storage versus privacy versus computational costs in class-incremental learning [C].We will add a discussion of this into the related work section.
>
> [C] Computationally Budgeted Continual Learning: What Does Matter? (CVPR 2023), Prabhu et al.
>
> ---
>
> > This paper, and many others like it, seek a CL method that trains a network from a random initial state. However, in practical industry settings, it's nearly always better to consider transfer learning, which for an image classifier has for a long time meant starting with a model trained on ImageNet, but nowadays is also meaning self-supervised pre-training. It's puzzling why the CL community ignore this for the most part. This might be changing now, with a bunch of papers in 2023 now doing CIL with pretrained ViT models (and also ResNets - e.g. your cited ref Panos et al). It could be a simple experiment to try this and assess the benefits of using your method in this different assumption. I am raising to highlight a potential opportunity, and to recommend you at least mention your assumption of the need to train a model from scratch rather than use pre-trained weights.
>
> We also agree with this point -- in fact, we always felt that Warm Start scenarios for exemplar-free CIL are a sort of simulation of starting from a pre-trained backbone. We ran an experiment starting from a backbone pre-trained on ImageNet-1K and then we apply EFCIL methods  on CIFAR-100, splitting the dataset into 20 equal splits. The table below gives these result (averaged over 5 runs).
>
> **CIFAR-100 Equal Split 20-step (from pretrained backbone)**:
> | Method | $A^K_{step}$ | $A^K_{inc}$|
> |---|---|---|
> LwF    | $22.48	 \pm 1.47$    |  $48.30\pm1.72$
> SSRE   | $33.66 \pm  2.30$    | $46.86 \pm 2.33$
> FeTrIL | $48.22	 \pm  0.47$   |  $62.30 \pm 0.91$
> EFC    | $\textbf{56.62}  \pm  0.30$   |   $\textbf{68.81} \pm   0.85$
>
> These results show that, even when starting from a strong backbone, the additional plasticity gained by EFC yields significant improvements over FeTrIL (which freezes the backbone).
>
> ---
>
> > As per above, do ablations on all datasets show the same trend, i.e. that the EFM+PR-ACE loss is enough to surpass comparison methods by itself, and adding prototypes rehearsal to this creates a further boost?
>
> See our comments on the first point above (and the new Appendix J). For all three datasets each component yields a noticeable boost in performance.
>
> ----
>
> > I found it very difficult to read the tiny font in Figure 2 and Figure 4(a), so please can you find a way to make those more easily readable?
>
> We increased the fonts size a little in these two figures and will make more refinements when preparing the final version.

---

> > ### Comment · Reviewer_sY5k · 2023-11-20
> >
> > Thankyou for the responses. It may be that there are some comparisons in the literature you can make for the pretrained backbone results stated above. Most of the work in this area now uses ViT backbones, but there is at least one recent paper that uses ResNets: https://arxiv.org/abs/2303.13199
> >
> > I see in this paper they use multiple different pre-trained backbones, one of which is ResNet 18. But I was unable in a short time to decide whether this paper provides any direct comparisons with your 20 task CIFAR-100 case.

---

### Official Review · Reviewer_4Tis · 2023-11-01

**Soundness:** 3 good
**Presentation:** 3 good
**Contribution:** 3 good
**Rating:** 6
**Confidence:** 5

**Summary:**

This article introduces a new EFCIL approach, called Elastic Feature Consolidation (EFC), which regulates changes in feature space directions most pertinent to previously-learned tasks while permitting greater adaptability in other directions. An essential contribution of this research is the establishment of a pseudo-metric in feature space generated by a matrix termed the Empirical Feature Matrix (EFM). Unlike the Fisher Information Matrix, the EFM can be easily stored and computed as it is independent of the number of model parameters, relying solely on the dimensionality of the feature space. To tackle the drift of the more flexible backbone, it also proposes an Asymmetric Prototype Replay loss (PR-ACE) that strikes a balance between new-task data and Gaussian prototypes during EFCIL. Lastly, it presents an enhanced method for updating class prototypes that leverages the already-computed EFM.

**Strengths:**

1- This paper presents a compelling and complex issue in incremental learning. Many current methods depend on storing exemplars from the previous task and require thorough exploration of highly informative exemplars to combat forgetting effectively. In contrast to existing approaches, the method proposed in this paper loosens the constraints regarding the availability of examples from the previous task. Consequently, it addresses a more practical and formidable problem.

2- The proposed method underwent experiments on three datasets, demonstrating a noteworthy enhancement in comparison to existing approaches. To substantiate the assertions and efficacy of the model components, a comprehensive ablation analysis was conducted and included in the paper.

**Weaknesses:**

1- In this method, each task is divided into disjoint classes, and the class boundaries are known for each task. However, can this method be extended to cases where the task boundaries are unknown for each task?

2- This method calculates the Empirical Feature Matrix (EFM), which determines the significant feature direction from a prior task, the mechanism of which remains unclear. Could you please provide a comprehensive explanation for better comprehension?
3- In Table 2, the results for the cold start issue demonstrate undesirable outcomes. Could you please provide an explanation in the main paper to justify this?

**Questions:**

Please address the questions raised in the weaknesses section.

---

> ### Author Response · Authors · 2023-11-18
>
> Many thanks to the reviewer for their kind words and probing questions. We agree that the exemplar-free scenario is indeed more practical (and more difficult) task. We take this opportunity to reply to specific questions raised by the reviewer.
>
> ---
> 1-Our Elastic Feature Consolidation (EFC) is designed for the *offline* class-incremental learning in which the full dataset is divided into multiple tasks and multiple epochs are performed during each training session. In this scenario *offline* implicitly assumes knowledge of task boundaries at training time, since whenever a new set of data arrives it is interpreted as a new incoming task. During inference however, the task boundary is not available. For EFC, knowledge of task boundaries is essential both to compute the Empirical Feature Matrix and to save the "previous model" to be used in the regularization term.
>
> To the best of our knowledge the existing literature on offline continual learning rarely addresses scenarios involving class overlap incremental sessions. A similar scenario was recently explored in [A] in which overlapping occurs between classes and each training session may include either new examples or a combination of old and new classes. However, even this setting requires task boundaries for transitioning from one session to the next. In theory, EFC could be applied to this benchmark and we find the idea of applying our method in such scenarios to be intriguing.
>
> Task-free incremental learning [B], in which there are no task boundaries for either training or inference, is more commonly addressed in online continual learning (which considers a single pass over all data). Adapting our method to this context is not trivial, as EFC requires multiple forward passes through the network (for the EFM computation at the end of training). We think that adapting EFC for task-free online settings, although non-trivial, might be possible using moving averages of models used to compute the EFM, and we may consider this possibility for future work.
>
> [A] General Incremental Learning with Domain-aware Categorical Representations (CVPR22), Xie et al.
>
> [B] Task-Free Continual Learning (CVPR19), Aljundi et al.
>
> ---
>  2-Computing the EFM after each training session uses the training images of the current task as input to the network. Its computation involves two calculations:
> * The *local feature matrix* per sample (Eq.5, Eq.7), computed using the derivative of log output probabilities wrt features.
> * The expectation of the local feature matrices over the current train set (Eq.6).
>
> The Empirical Feature Matrix (EFM) identifies feature directions important to prior tasks. Consider the EFM $E_t$, computed after training task $t$, and its eigenvector decomposition:
> $$
> E_t = U_t^T \Lambda_t U_t,
> $$
> where $U_t, \Lambda_t$ represents the eigenvector and eigenvalues of $E_t$. The eigenvectors with non-zero eigenvalues represent the *principal directions* for the prior tasks. By construction, applying perturbations in these directions the performance of the previous-task classifiers degrade (Fig.2). Conversely, the *non-principal* directions are associated with zero eigenvalues and perturbing along them does not have any impact on model output. This behavior is justified by its interpretation in terms of KL-divergence (Eq.8).
>
> We exploit this property by using the EFM into the regularization loss (Eq. 9).  Let's focus on the term in which it appears and consider its spectral decomposition. Using the paper notation and defining $\delta(x)=f_t(x) - f_{t-1}(x)$ to denote the feature drift for $x \in \mathcal{X}_t$, we can express the regularization term as:
>
> $$
> \delta(x)^T  E_{t-1} \delta(x) = (U_{t-1} \delta(x))^T  \Lambda_{t-1} (U_{t-1} \delta(x))
> $$
>
> The plasticity in training is given by the diagonal matrix containing the eigenvalues of the EFM. In particular, we regularize feature drift in principal directions of the EFM with a strength determined by its eigenvalues.
>
> ---
> 3-The results in Table 2 refer to the memory-efficient variants of our method. In the Warm Start scenario, storing one covariance per class has a significant impact since the covariances remain fixed. With respect to the Cold start(CS) scenario in this specific setting, the feature drift is mitigated by the fact that we learn half of all the classes during the first task. On the contrary, in CS, the representations are more prone to changes, and having per-class fixed covariances is less crucial. Consequently, the differences in CS Scenario between the variants of our method are practically negligible. Theoretically, both scenarios, especially CS, could be improved if we design a method to update these matrices during the incremental steps. However, as we highlight in the limitations paragraph, this remains an open point.
>
> ---
> In the final version of the paper, we will integrate elements of the above discussion to improve understanding of the EFM and its use for regularizing incremental learning.

---

### Author Response · Authors · 2023-11-20
**Author Rebuttal by Authors**

We reiterate our appreciation to all reviewers for their insightful comments aimed at improving the quality of our work. In this brief, final message we summarize our responses and new results presented during the discussion period.

In response to the suggestions provided by [Reviewer 4TiS](https://openreview.net/forum?id=7D9X2cFnt1&noteId=MhHUSBZeWV), we have included additional explanations on EFC and explored other potential applicative task-free scenarios for EFC.

As suggested by [Reviewer sY5k](https://openreview.net/forum?id=7D9X2cFnt1&noteId=1rtjuKlk9C), we have conducted a more comprehensive ablation analysis on our approach, extending all ablations to Tiny-ImageNet and Imagenet-Subset in both Cold Start and Warm Start scenarios (refer to the new Appendix J). Additionally, we applied EFC in a different scenario for class-incremental learning, one starting from a pre-trained backbone.

Addressing the concerns raised by [Reviewer Hr2j](https://openreview.net/forum?id=7D9X2cFnt1&noteId=iVEmtgVd9y), we evaluated EFC and compared it with FeTrIL on ImageNet-1K, a class-incremental learning scenario involving a substantial amount of data and classes. These experiments, which we performed in both Warm and Cold Start scenarios, illustrate the versatility of EFC and show improved performance, particularly in the Cold Start scenario, when compared to our nearest competitor.

For [Reviewer RQMN](https://openreview.net/forum?id=7D9X2cFnt1&noteId=5mRtetyP0R), we expanded the discussion on Cold Start versus Warm Start and conducted [additional experiments](https://openreview.net/forum?id=7D9X2cFnt1&noteId=LW3xYn6Xl0) to explore the regularization impact of the EFM (see the new appendix K). Recognizing the nuanced nature of the Cold Start vs. Warm Start discussion, we plan to incorporate this additional discussion in the final version of the paper.

All these suggestions will be taken into account (and new results incorporated) when preparing the final version of our paper, and we again thank everyone for their constructive criticism of our contribution.

---

### Meta-Review · Area_Chair_qam6 · 2023-12-09

**Metareview:**

In this paper, the authors propose an exemplar-free class-incremental learning method that regularizes changes in directions in feature space most relevant for previously learned tasks and allows more plasticity in other directions. Extensive experimental results are provided in this paper. After the rebuttal session, most of the concerns were addressed (e.g., ImageNet-1k experiments), and all reviewers are positive towards this paper. Therefore, I recommend acceptance. The authors need to incorporate the additional results and discussions during the rebuttal session into the final version.

**Justification For Why Not Higher Score:**

Even though this is a good paper, there are still some small weaknesses. For example, some popular continual learning methods are not directly compared. ImageNet-1k experiments are not fully completed. Therefore, the final recommendation is a poster.

**Justification For Why Not Lower Score:**

This is a clear acceptance. All reviewers agree to accept this paper after the rebuttal session.

---

### Decision · Program_Chairs · 2024-01-16

Accept (poster)